# Structure and Productivity of the Phytoplankton Community in the Southwestern Kara Sea in Early Summer

Sergey A. Mosharov [1,2,*], Elena I. Druzhkova [3], Andrey F. Sazhin [1], Pavel V. Khlebopashev [1], Anastasia N. Drozdova [1], Nikolay A. Belyaev [1] and Andrey I. Azovsky [1,4]

1   Shirshov Institute of Oceanology, Russian Academy of Sciences, 117997 Moscow, Russia
2   Faculty of Power Engineering, Bauman Moscow State Technical University, 105005 Moscow, Russia
3   Murmansk Marine Biological Institute of Russian Academy of Sciences, 183010 Murmansk, Russia
4   Biology Faculty, Lomonosov Moscow State University, 119991 Moscow, Russia
*   Correspondence: sampost@list.ru

**Abstract:** Knowledge of the features of the structure and productivity of the Arctic communities of marine planktonic algae is necessary to identify possible changes in the pelagic ecosystem functioning under the changing climate condition of the Kara Sea. This study shows that the species diversity, abundance of phytoplankton, and production activity of algae are at a maximum at the beginning of summer during a seasonal ice melting period. The studies were carried out in the southwestern Kara Sea and in the estuarine zone of the Ob and Yenisei rivers from 29 June to 15 July 2018. The concentrations of nutrients and dissolved organic carbon were determined. The optical properties of chromophoric dissolved organic matter, species composition, abundance and biomass of all size groups of phototrophic and heterotrophic phytoplankton, and parameters of primary production and potential photosynthetic capacity were considered. Statistical data analysis showed that the leading factors influencing changes in the abundance of phytoplankton and its productivity are the content of silicates and salinity. At the same time, the production potential of algae is realized as short-lived and small phytoplankton assemblages differed in number taxa and diversity, with an equally rapid decrease in photosynthetic activity. Such changes affect the Marine Zone to a greater extent and the Estuarine Zone to a lesser extent.

**Keywords:** phytoplankton; primary production; chlorophyll; freshwater runoff; nutrients; Kara Sea

## 1. Introduction

Knowledge of the structural features of phytoplankton and an understanding of how Arctic communities of marine planktonic algae currently function are necessary to identify possible changes in the vital activity of the pelagic ecosystem under conditions of climate change and an increasing anthropogenic load on the Kara Sea. The Kara Sea is a heterogeneous and productive marine ecosystem in the Arctic Ocean that plays a key role in the carbon cycle. It is mostly a shallow Arctic shelf basin influenced by river runoff. The Kara Sea receives 1300–1400 km$^3$ of fresh water annually, which accounts for 41% of the total freshwater runoff to the Arctic Ocean [1]. Coastal marine systems in the Arctic typically contain high concentrations of inorganic and organic particles, which enter the water column via the melting of land and sea ice [2]. Studies conducted in different Arctic regions have demonstrated that all components of the plankton community play a significant role in the functioning in cold waters, particularly in the southwestern Kara Sea [3,4].

In recent years, data on phytoplankton abundance and their production characteristics on the Kara Sea shelf in early spring, late summer, and autumn have been obtained [3,5–14]. Additionally, the species composition of all size groups of algae, phytoplankton abundance, and their production characteristics in the Kara Sea in the phenological period of early

summer remains unstudied. The only exception is [14], which considered the values of primary production and chlorophyll *a* (Chl *a*) during a period of seasonal ice melting in the northwestern Kara Sea. The species composition of algae was not presented, except for six generic taxa of diatoms predominant in plankton. This study also provides no data on the Chl *a* concentration and measured primary production on the sea surface. Only the calculated integral values of these indicators in the water column are given.

A feature of the phenological period between spring and summer is that, in the western part of the sea, planktonic algae communities function at this time under ice melting conditions. In the eastern Kara Sea, in early summer, phytoplankton are under pressure by the maximum influx of water masses from the Ob and Yenisei rivers, which alter the salinity regime of this area and transport a huge amount of detrital particles. All of this (mainly salinity and silicate) affects not only the species composition and abundance of diatoms, but also the functioning and production characteristics of all groups of phytoplankton. The results of our research make it possible to fill in the missing information on the state of planktonic algae, their species diversity, and their production characteristics during the early summer. Thus, it is possible to partially reconstruct the seasonal cycle of plankton communities in two water areas of the Arctic shelf, unaffected and affected by freshwater runoff.

The hypothesis we want to test based on this study is that the species diversity and phytoplankton abundance are at a maximum precisely in early summer, not in other seasons (in spring; in mid-summer, when the flood of the Ob and Yenisei ends; or later, during the rivers' autumn flood). In addition, we assume that algae activity reaches its maximum values (albeit for a short time) precisely in early summer and the primary production values rapidly increase on the sea surface. The high production potential of planktonic algae can be realized as a mosaic of small, short-lived water areas with different structural organization of the phytoplankton assemblages within the period of seasonal ice melting. These assemblages differed mostly in number of taxa and diversity.

The aim of this work is to assess phytoplankton abundance and activity in the Kara Sea in early summer in and outside the river runoff zone of influence when the water area is freed from ice cover. The specific objectives were to determine (1) the abundance of phototrophic and heterotrophic planktonic algae, (2) the species composition of phytoplankton and the relationship between species, (3) the Chl *a* concentration, (4) the rate of primary production and photosynthetically determined potential photosynthetic capacity of phytoplankton, and (5) the main environmental variables governing the phytoplankton's variability.

## 2. Materials and Methods

### 2.1. Study Sites and Sampling

Water samples were collected from 29 June to 15 July 2018 on board the '*Norilskiy Nickel*' at 25 stations along the vessel's course from a station in the Barents Sea, near the Kara Strait, to a station near the Taimyr Peninsula in the Yenisei estuary (stage 1, June 29–July 1) and back to the Kara Strait (stage 2, 12–15 July) (Figure 1). Stage 1 included work at stations 1–14; stage 2, at stations 15–25. Stations were located in an area of the shelf area that receives no river runoff, the marine area, and in the shelf area adjacent to the Ob and Yenisei estuaries, the coastal area. As the vessel moved, surface water samples were collected to determine the hydrophysical (temperature and salinity), hydrochemical (alkalinity, concentrations of nitrite+nitrate, phosphate, silicate, and dissolved organic carbon), and hydrobiological (species composition, abundance and biomass phytoplankton, Chl *a* concentration, chlorophyll fluorescence, and primary production) variables. The degree of ice coverage (sea ice concentration and SIC) was assessed visually from the ship.

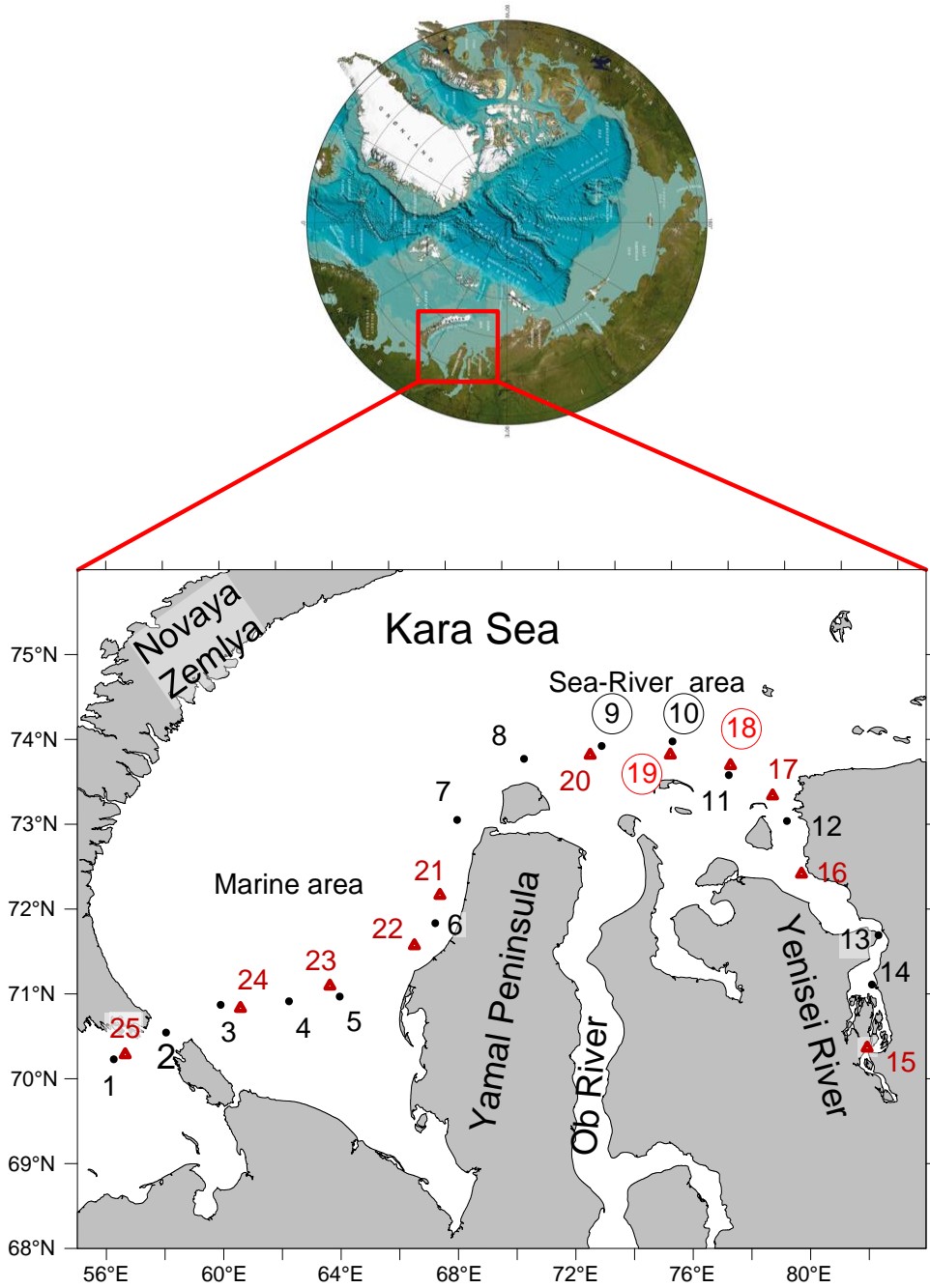

**Figure 1.** Location of sampling stations in the Kara Sea: circles, stage 1; triangles, stage 2. Station numbers in circles, frontal zone.

Temperature was measured with an SBE-39 probe and LCD-thermometer HI98509 Checktemp-1 (HANNA Instruments, Вунсокете, Woonsocket, RI, USA). Salinity was measured with a Kelilong PHT-028 salinity meter (Kelilong Electron, Fuan, China). Surface water samples (depth 0.5 m) for biological variables and dissolved organic carbon (DOC) were collected manually with sterile 10 L buckets from the side of the ship. The 10 L buckets were rinsed with 0.1 M HCL each time before sampling.

Samples to determine alkalinity, concentrations of nitrite+nitrate, phosphate silicate, and DOC were collected in 0.5 L plastic bottles without preservation, treated immediately after sampling. For work in areas with a considerable quantity of particulate organic matter (POM) (bays and river–sea interfaces), the water samples were preliminarily passed through a 1 μm Nuclepore filter. The dissolved inorganic phosphorous (P-$PO_4$), dissolved inorganic

silicate ($Si(OH_4)$), nitrite nitrogen ($N-NO_2$), and nitrate nitrogen ($N-NO_3$) concentrations were measured by using standard procedures [15]. The DOC concentrations were measured with a Shimadzu TOC-Vcph carbon analyzer coupled with an SSM-5000A solid sample module (Shimadzu, Kyoto, Japan) [16].

The intensity of surface photosynthetically active radiation (PAR, 400–700 nm) was measured with an LI-190R quantum sensor and an LI-1400 data-logger (LI-COR, Lincoln, USA).

### 2.2. Optical Properties Chromophoric (Colored) Dissolved Organic Matter (CDOM)

The samples were passed through precombusted Whatman GF/F filters (GE Healthcare, Chicago, USA) with a pore size of ~0.7 μm. The filtrate was collected into acid-cleaned 30 mL glass vials and stored under dark conditions at 4 °C until further analysis. Absorption spectra ($A_\lambda$) were recorded at room temperature with a T80 spectrophotometer (PG Instruments, Leicestershire, UK) in 1 cm quartz cells. Measurements were performed within the spectral range of 220–700 nm in 2 nm increments. The blank-corrected absorption spectra were converted into Napierian absorption coefficients $a_\lambda$ with the following equation:

$$a_\lambda = \frac{2.303 A_\lambda}{l}$$

where $l$ is 0.01, the cell path length in meters.

Absorption spectra were recorded within the wavelength ranges 275–295 and 350–400 nm and were characterized by the exponential spectral slope coefficient $S$ as suggested by Stedmon and others [17,18],

$$a_\lambda = a_{\lambda_0} e^{-S(\lambda - \lambda_0)}$$

and denoted as $S_{uvb}$ and $S_{uva}$, respectively. The $S_{uvb}$ and $S_{uva}$ values were determined by linear regression of the log-transformed functions of absorption coefficients $a_\lambda$ [19]. The spectral slope ratio $S_r$ was calculated as follows:

$$S_r = \frac{S_{uvb}}{S_{uva}}$$

A Specific UV absorbance (SUVA) was calculated by normalizing the decadic absorption at 254 nm to the DOC concentration. It has proven to be a useful parameter for estimating the dissolved aromatic carbon content in aquatic systems [20].

Fluorescence measurements were taken with a Fluorat-02-Panorama spectrofluorometer (Lumex, Saint Petersburg, Russia) in a 1 cm quartz cell. Emission spectra were recorded between 240 and 650 nm with a 2 nm increment, and the excitation wavelengths were recorded from 230 to 550 nm with a 5 nm increment. The accuracy of excitation and detection wavelength settings was ascertained based on the Xe atomic line position and estimated as ±1 nm; the spectral resolution of the monochromators was 5 nm. All spectra were corrected for inner-filter effects. The humification index HIX was calculated as a ratio of integral fluorescence intensities

$$HIX = \frac{\sum_{435}^{480} I_\lambda}{\sum_{300}^{345} I_\lambda}$$

at an excitation wavelength of 254 nm, as suggested by Zsolnay et al. [21]. The biological/autochthonous index BIX is the fluorescence intensity ratio

$$BIX = \frac{I_{380}}{I_{430}}$$

at an excitation wavelength of 310 nm [22]. All the calculations were performed with the Albatross package [23].

### 2.3. Phytoplankton

To take into account pico-, nano- and microphytoplankton, as well as to determine their trophic status, 20 mL of a sample was stained by fluorochrome primulin, fixed with 3.6% glutaric dialdehyde solution, and deposited on black nuclear filters with a 0.4 μm pore diameter [24–26] using our own modification of the procedure [27]. Filters were placed on glass slides and covered with Leica Typ N immersion liquid and a cover glass. Immediately after preparation, the slides were frozen and stored at −24 °C. The slides were analyzed under stationary conditions with a Leica DM 5000B luminescence microscope (Leica Microsystems, Wetziar, Germany) at ×200–1000 magnification in two replicates. Small numerous forms were taken into account in 50–100 fields of vision; the rest, during complete examination of the preparation.

Duplicate samples with a volume of 500 mL and also fixed by 3.6% glutaric dialdehyde solution were treated in Nogeotte chambers (volume 0.045 mL). The samples were pre-liminarily concentrated with excess water removed through a tube covered by a $5 \times 5$ μm nylon mesh. The sample concentrate was examined completely under a Carl Zeiss Axio Imager D1 light microscope (Carl Zeiss Microscopy Deutschland GmbH, Jena, Germany) at ×400 magnification. The volume of cells was calculated based on the corresponding stereometric figures. The wet biomass was estimated based on the volume of an individual cell using Image Scope Color, version Image Scope M 2009, Fei Electron Optics BV. The algal biomasses were recalculated to carbon equivalent based on their volumes [28].

Algae nomenclature are given in accordance with AlgaeBase [29] and WoRMS [30].

### 2.4. Primary Productivity Parameters

Primary production (PP) was measured on board using the $^{14}$C uptake method [31]. The samples were incubated in polycarbonate flasks (50 mL) for 3 h in a laboratory incubator with individually adjustable LED lighting and temperature maintenance using a HAILEA-100 laboratory cooler (Hailea, Guangzhou, China). In the incubator, to simulate light conditions corresponding to the sampling depths, each flask was illuminated by an individually adjustable LED panel (white light) with the illumination level controlled with an LI-192SA quantum sensor (LI-COR, Lincoln, USA). The light flux of each LED panel was regulated by changing the current.

After incubation, flasks were filtered onto a 0.45 μm membrane (Vladipore, Vladimir, Russia). Radioactivity of the samples was determined with a Triathler (Hidex, Turku, Finland) liquid scintillation counter.

Biomass-specific PP, $P^B$ (mgC mgChl $a^{-1}$ $h^{-1}$) was calculated by normalizing PP at different depths to the corresponding Chl $a$ concentration.

The Chl $a$ concentration was measured fluorometrically [32]. Seawater samples (500 mL) were filtered onto Whatman GF/F fiberglass filters under low vacuum (<0.3 atm). For extraction, the Chl $a$ filters were placed in acetone (90%) and maintained at a temperature of +4 °C in darkness for 24 h. Then, the fluorescence of the extracts was measured with a MEGA-25 fluorometer (MSU, Moscow, Russia) [33] before and after acidification with 1 N HCl. The fluorometer was calibrated before and after each cruise using pure Chl $a$ (Sigma) as a standard. The Chl $a$ and phaeophytin $a$ concentrations were calculated according to [34].

Active Chl $a$ fluorescence was measured with a MEGA-25 PAM-fluorometer (MSU, Moscow, Russia) [33]. Prior to measurement, the samples were kept in the dark for at least 20 min [35]. The minimum ($F_0$) and maximum ($F_m$) fluorescence of the samples was measured. The maximum quantum efficiency of PSII ($F_v/F_m$) was calculated as [36]:

$$F_v/F_m = (F_m - F_0)/F_m.$$

The $F_v/F_m$ value indicates the potential photosynthetic capacity of phytoplankton. The maximum $F_v/Fm$ values for phytoplankton under optimal conditions correspond to 0.70, with a significant difference between taxa [37,38].

### 2.5. Statistical analyses

The species richness index (number of species/taxa), species diversity index (Shannon' index H to base e) and community evenness (Pielou's index J) were calculated for each station according to biomass data. To test the differences in the central tendency between subsets of univariable data, the nonparametric Mann–Whitney pairwise median test was applied. To analyze compositional changes in communities among stations, we used nonmetric multidimensional scaling (*n*MDS-plot) based on Bray–Curtis similarity.

Multivariate nonparametric distance-based regression analysis (DistLM) was used to explore the relationships between variations in the phytoplankton characteristics (biomass, production, diversity, and composition) and eight environmental parameters (stage, water temperature, salinity, ice coverage, phosphate, silicate, nitrate, and nitrite). The stepwise selection procedure with an adjusted $R^2$ criterion for including/excluding a predictor into the best-fitting model was applied. Euclidean distance was used as a dissimilarity measure for univariate response variables, and Bray–Curtis similarity for species composition analysis [39]. Nonparametric tests were used for pairwise comparisons (Mann–Whitney median test for univariate data, ANOSIM test for multivariate data). Statistical analyses were performed using PRIMER 7 with the PERMANOVA+ add-on (PRIMER-E Ltd., Plymouth, UK) and PAST 4.12 software [40].

## 3. Results

### 3.1. Environmental Parameters

Based on the salinity values of the surface water layer, three distinct zones were identified in the study area: the marine zone (MZ) to the west of the Yamal Peninsula, unaffected by freshwater runoff and with a salinity of 26.9–33.4 psu (stations 2–8 at stage one and stations 21–24 at stage two); the frontal zone (FZ), with a salinity of 11.0–15.5 psu (stations 9 and 10 at stage one and stations 18 and 19 at stage two); and the estuarine zone (EZ), with a salinity of 0–2.2 psu (stations 11–14 at stage one and stations 15–17 at stage two) (Figure 2). It should be noted that at stage one (end of June), the MZ–FZ boundary was quite pronounced in terms of salinity values: at stations eight and nine, the salinity values differed by a factor of two, which is also confirmed by the SUVA values, which showed the boundary between the predominance of autochthonous and allochthonous (terrestrial dissolved derived organic matter) suspended particulate matter (SPM) in water (Figure 3).

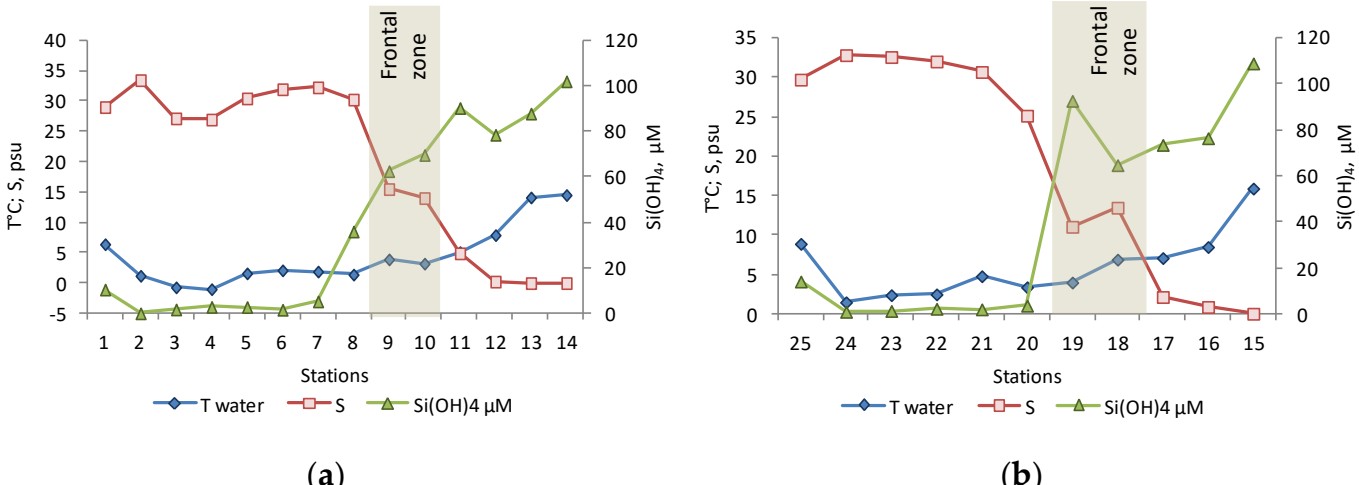

**Figure 2.** Distribution of surface temperature (T, °C), salinity (S, psu), and silicate concentration (Si(OH)$_4$, μM) in the study area at stage 1 (**a**) and stage 2 (**b**).

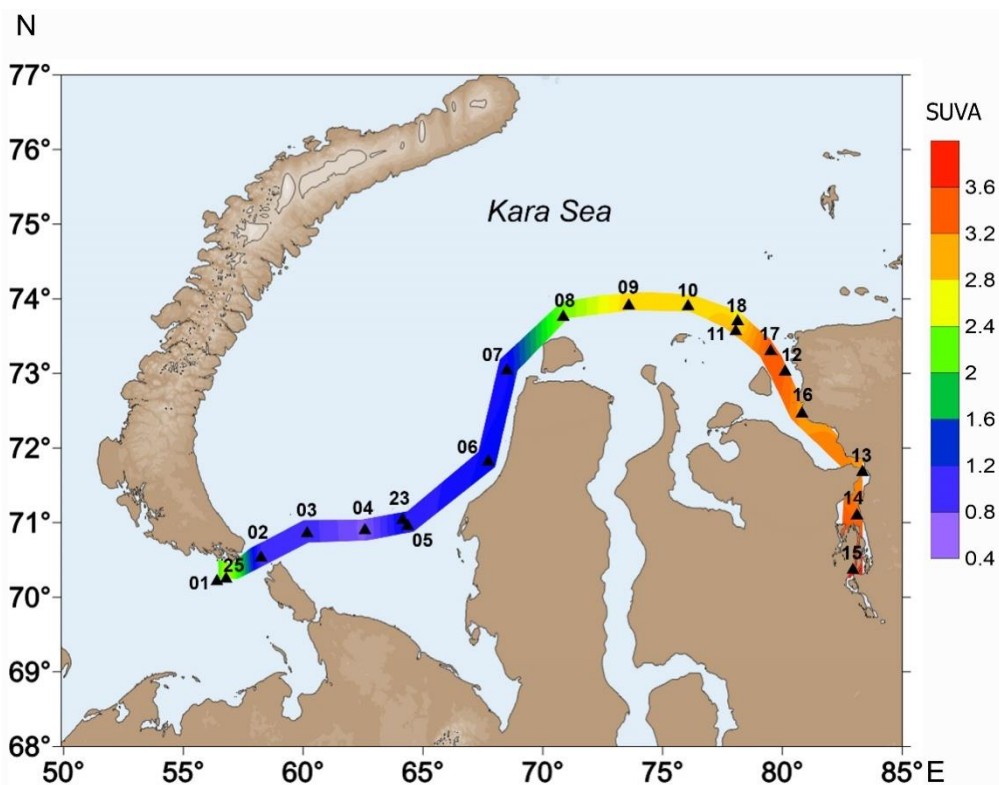

**Figure 3.** Variation of specific UV absorbance (SUVA, m²/gC) at stages 1 and 2.

At stage two (mid-July) in the same area (stations 19 and 20), the salinity boundary was less pronounced (Figure 2). The silicate concentration, which serves as an indicator of the influence of river runoff, in the MZ was at a low level (2.27 ± 1.40 μM at stations 2–7 and 20–24), but then increased in the FZ and EZ (35.9–108.8 μM). Additionally, in contrast to salinity, the silicate concentration at the MZ–FZ boundary at stage one increased gradually, while at stage two, it increased very sharply (Figure 2).

The surface water temperature in the MZ at stage one (end of June) ranged from −1 to +2.1 °C, while at stations three and four, located within a continuous ice field, the water temperature was negative (from −0.7 to −1 °C). It should be noted that the surface salinity at these stations was also significantly lower than that at other stations in the MZ (27 psu). At stage two (mid-July), the surface temperature in this area was slightly higher (from +1.5 to + 4.8 °C). In the freshened area (from the FZ to EZ), the temperature gradually increased from +5.1 to 14.5 °C at stage one and from +7.1 to 15.9 °C at stage two (Figure 2).

The content of dissolved inorganic nitrogen (DIN = $NO_3$ + $NO_2$) in the MZ at stage one was 0.07–4.34 μM, significantly increasing in the central part to 3.03 μM (station four) and near the FZ up to 4.34 μM (station eight) (Table 1). Two weeks later, at stage two, the DIN values in this zone were at a very low level of 0.06–0.68 μM. In the FZ, DIN values sharply increased on seaward (stations 9 and 19) and significantly decreased on the estuarine side (stations 10 and 18) at both stages. In the freshened zone and at the beginning of the EZ, at stage one, the DIN values again increased to 3.87–7.24 μM, then decreased in river water to 1.22–1.34 μM. At stage two, in the low salinity zone, the DIN values did not exceed 0.39 μM.

The phosphate content at stage one changed insignificantly in the MZ from 0.07 to 0.24 μM, significantly varied in the FZ from 1.01 to 0.21 μM, and increased in the low salinity zone to 0.46–0.91 μM. At stage two, the phosphate concentration distribution was more uniform, with low values in the MZ (0.07–0.13 μM) and a moderate increase in the FZ and EZ (up to 0.20–0.46 μM) (Table 2).

**Table 1.** Physical properties of the sampling locations in the Barents and Kara seas from June 29 to July 15, 2018: longitude (Long) and latitude (Lat), sea ice concentration (SIC), surface temperature (T), salinity (S).

| Area, Stage | Station | Long, E,° | Lat, N,° | Date | SIC,% | T, °C | S, psu |
|---|---|---|---|---|---|---|---|
| Barents Sea, Stage 1 | 1 | 56°17.6 | 70°10.9 | 29.06.2018 | 0 | 6.4 | 28.98 |
| Barents Sea, Stage 2 | 25 | 56°40.8 | 70°12.9 | 15.07.2018 | 0 | 8.9 | 29.70 |
| Kara Strait, Stage 1 | 2 | 58°08.8 | 70°29.9 | 29.06.2018 | 0 | 1.2 | 33.42 |
| Kara Sea, Marine Zone, Stage 1 | 3 | 60°04.8 | 70°49.6 | 29.06.2018 | 100 | −0.7 | 27.12 |
| | 4 | 62°30.1 | 70°52.1 | 29.06.2018 | 100 | −1.0 | 26.93 |
| | 5 | 64°17.8 | 70°55.5 | 29.06.2018 | 25 | 1.6 | 30.41 |
| | 6 | 67°40.8 | 71°47.5 | 29.06.2018 | 0 | 2.1 | 31.92 |
| | 7 | 68°27.7 | 73°00.9 | 30.06.2018 | 0 | 1.9 | 32.23 |
| | 8 | 70°49.4 | 73°44.2 | 30.06.2018 | 0 | 1.4 | 30.22 |
| Kara Sea, Marine Zone, Stage 2 | 20 | 73°06.6 | 73°47.4 | 14.07.2018 | 0 | 3.4 | 25.13 |
| | 21 | 67°50.8 | 72°08.3 | 14.07.2018 | 0 | 4.8 | 30.67 |
| | 22 | 66°55.4 | 71°32.9 | 14.07.2018 | 0 | 2.5 | 32.00 |
| | 23 | 64°04.9 | 71°00.5 | 14.07.2018 | 25 | 2.4 | 32.56 |
| | 24 | 60°46.9 | 70°48.1 | 15.07.2018 | 25 | 1.5 | 32.80 |
| Kara Sea, Frontal Zone, Stage 1 | 9 | 73°34.4 | 73°53.3 | 30.06.2018 | 0 | 3.9 | 15.48 |
| | 10 | 76°03.6 | 73°52.7 | 30.06.2018 | 0 | 3.2 | 14.00 |
| Kara Sea, Frontal Zone, Stage 2 | 18 | 78°08.4 | 73°40.8 | 13.07.2018 | 0 | 6.9 | 13.48 |
| | 19 | 76°00.1 | 73°47.9 | 13.07.2018 | 0 | 4.0 | 11.06 |
| Estuarine Zone, Stage 1 | 11 | 78°03.6 | 73°32.5 | 30.06.2018 | 0 | 5.1 | 4.90 |
| | 12 | 80°08.1 | 73°00.1 | 01.07.2018 | 0 | 7.9 | 0.25 |
| | 13 | 83°22.7 | 71°39.1 | 01.07.2018 | 0 | 14.1 | 0.001 |
| | 14 | 83°08.6 | 71°04.0 | 01.07.2018 | 0 | 14.5 | 0.001 |
| Estuarine Zone, Stage 2 | 15 | 82°58.5 | 70°19.9 | 12.07.2018 | 0 | 15.9 | 0.05 |
| | 16 | 80°50.4 | 72°26.4 | 13.07.2018 | 0 | 8.5 | 0.91 |
| | 17 | 79°31.2 | 73°16.2 | 13.07.2018 | 0 | 7.1 | 2.17 |

**Table 2.** Chemical properties of the sampling locations in the Barents and Kara Seas from 29 June to 15 July 2018: alkalinity (Alk), dissolved inorganic nitrogen concentration (DIN = $NO_2$ + $NO_3$), phosphate concentration $PO_4$, silicate concentration (Si), and concentration of dissolved organic carbon (DOC).

| Area, Stage | Station | Alk, mg-eq $L^{-1}$ | DIN, μM | $PO_4$, μM | Si, μM | DOC, mg $L^{-1}$ |
|---|---|---|---|---|---|---|
| Barents Sea, Stage 1 | 1 | 2.312 | 0.21 | 0.09 | 10.51 | 2.62 |
| Barents Sea, Stage 2 | 25 | ND * | 1.59 | 0.06 | 14.16 | 2.92 |
| Kara Strait, Stage 1 | 2 | ND | 0.95 | 0.07 | 0.06 | 1.42 |
| Kara Sea, Marine Zone, Stage 1 | 3 | 1.789 | 1.56 | 0.10 | 1.76 | 1.39 |
| | 4 | 1.781 | 3.03 | 0.14 | 2.89 | 2.34 |
| | 5 | 2.274 | 0.44 | 0.19 | 2.83 | 1.66 |
| | 6 | 2.228 | 0.07 | 0.20 | 1.64 | 2.11 |
| | 7 | 2.266 | 0.11 | 0.24 | 5.29 | 2.38 |
| | 8 | 2.001 | 4.35 | 0.52 | 35.99 | 3.94 |

**Table 2.** *Cont.*

| Area, Stage | Station | Alk, mg-eq L$^{-1}$ | DIN, µM | PO$_4$, µM | Si, µM | DOC, mg L$^{-1}$ |
|---|---|---|---|---|---|---|
| Kara Sea, Marine Zone, Stage 2 | 20 | 1.531 | 0.68 | 0.26 | 3.66 | 6.96 |
| | 21 | 1.933 | 0.11 | 0.07 | 2.08 | 3.30 |
| | 22 | 1.758 | 0.06 | 0.10 | 2.33 | 2.13 |
| | 23 | 2.031 | 0.06 | 0.07 | 1.38 | 1.98 |
| | 24 | ND | 0.09 | 0.13 | 1.01 | 1.62 |
| Kara Sea, Frontal Zone, Stage 1 | 9 | 1.47 | 5.91 | 1.01 | 62.48 | 7.93 |
| | 10 | 1.637 | 2.09 | 0.21 | 69.59 | 5.87 |
| Kara Sea, Frontal Zone, Stage 2 | 18 | 1.385 | 0.17 | 0.33 | 64.68 | 7.85 |
| | 19 | 1.289 | 7.95 | 0.37 | 92.62 | 9.36 |
| Estuarine Zone, Stage 1 | 11 | 1.342 | 3.87 | 0.46 | 90.23 | 11.46 |
| | 12 | 0.462 | 7.24 | 0.91 | 78.46 | 9.64 |
| | 13 | ND | 1.22 | 0.58 | 87.77 | ND |
| | 14 | ND | 1.34 | 0.80 | 101.93 | ND |
| Estuarine Zone, Stage 2 | 15 | ND | 0.60 | 0.34 | 108.85 | ND |
| | 16 | ND | 0.99 | 0.46 | 76.38 | ND |
| | 17 | 0.644 | 0.35 | 0.20 | 73.49 | 9.48 |

ND *—no data.

### 3.2. CDOM Optical Properties

The optical indices calculated from the absorption and fluorescence spectra suggest considerable variation in the nature of chromophoric dissolved organic matter (CDOM) in the studied samples. As mentioned earlier, the significant impact of terrestrial-derived dissolved organic matter (DOM) was observed only east of the Yamal Peninsula (stations 8–18, Figure 3). The humification index (HIX) varied between 5.6 and 12.3, with the maximum values observed in the Yenisei River estuary. The average values of biological/autochthonous index (BIX), spectral slope ratio (Sr), and SUVA in this region were 0.6, 0.9, and 3.0 m$^2$/gC). The maximum SUVA value was measured in the Yenisei River estuary (3.79 m$^2$/gC at stations 15–18). With increasing distance from the Yenisei River estuary, SUVA dropped gradually (Figure 3). In the riverine waters and river-influenced ocean margins, the SUVA spectral slope ratio varied between 13.4 and 18.1 µm$^{-1}$. West of the Yamal Peninsula, the influence of Ob and Yenisei runoff was weak. The humification index (HIX) did not exceed 3.9 and SUVA varied between 0.56 and 1.72 m$^2$/gC.

### 3.3. Structure of the Phytoplankton Community

In the studied water area of the southwestern Kara Sea, 89 species and supraspecific taxa of planktonic algae were recorded over the entire research period, most of which (65%) belong to the Bacillariophyta division. Among the small flagellar forms, the most common were *Dicrateria rotunda*, *Plagioselmis prolonga*, the haploid form of the diploid *Teleaulax amphioxeia*, and *Pyramimonas* spp.

Stage one. In the period from June 29 to July 1, the basis of the phytoplankton community in the MZ was an assemblage of centric diatoms of the genus *Thalassiosira* (*T. hyalina*, *T. decipiens*, *T. gravida*, *T. antarctica var. borealis*, *Thalassiosira* spp.), as well as colonial forms (*Pauliella taeniata*, *Navicula septentrionalis*, *Pseudo-nitzschia seriata f. seriata*) and solitary (*Nitzschia longissima*) representatives of pennate diatoms. There were numerous phototrophic (*Peridiniella catenata*) and heterotrophic (*Protoperidinium bipes*) Dinophyceae and Euglenoidea algae (*Eutreptiella braarudii*, *Eutreptiella* sp.), as well as the phototrophic ciliate *Myrionecta rubra*. These species were present at almost all stations, except for river ones, constituting the main abundance and biomass of phytoplankton. The abundance of *Thalassiosira* spp. ranged from 9 to 86% of phytoplankton abundance, and in most cases exceeded 70%. The abundance of *Pauliella taeniata* varied within 7–84% of the total

concentration of planktonic algae. The total abundance of phytoplankton at stage one varied within 153–1528 × $10^3$ cells/L; the algae biomass varied from 2.78 to 230.72 mgC/m$^3$ (Figure 4). The role of small flagellate algae was significant only at stations one–four, 22–99%, and at stations five and six, respectively, 9.2 and 1.4% of the total phytoplankton biomass. At other stations at stage one, the value did not exceed 1.8% of the total algae biomass. Under continuous ice cover (stations three and four), under the ice, in addition to the basic assemblage of species, the development of cryoflora was observed, namely *Nitzschia frigida* and *Melosira arctica var. bornholmiensis*, which account for 26–27% of the total phytoplankton abundance and 5–6% of the total algae biomass. At station five, the ice cover was only 25% of the water area and both species associated with ice completely vanished from the plankton. The leading position in the community was occupied by *Chaetoceros* spp. (80% of the total phytoplankton abundance or 29% of total biomass). Additionally, the basic species assemblage continued to develop actively; the total abundance of the community increased by nine times, and that of biomass by two times versus earlier stations. At station six, the share of *Chaetoceros* spp. gradually decreased against the development of the main species assemblage, which yielded a further increase in quantitative indicators: the number of algae increased by 2.2 times, and that of biomass by 2.7 times.

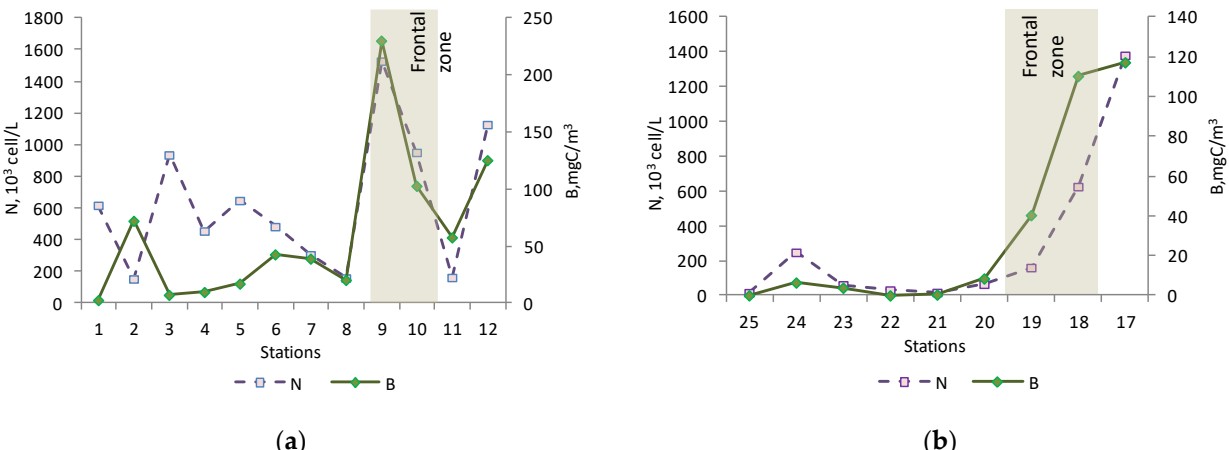

**Figure 4.** Distribution of abundance (N) and biomass (B) of phytoplankton in the study area at stage 1 (**a**) and stage 2 (**b**).

In the west of the MZ (station seven), significant structural changes were observed in the phytoplankton community. The growth in the number of representatives of the genus *Thalassiosira* continued, while some species passed into the spore formation stage. The relative abundance of *Chaetoceros* spp. decreased to 17%, and that of *Pauliella taeniat*, to 8% of the total phytoplankton abundance. Additionally, the assemblage of early spring colonial pennate diatoms (*Fragilariopsis oceanica*, *Navicula septentrionalis*) actively developed, which almost completely vanished from plankton further east. The change in species composition led to a decrease in the total number of algae by 1.6 times at almost the same level of biomass.

The water area of the eastern part of the FZ (station nine) was characterized by the maximum development of the basic assemblage of algae species throughout the entire observation period with dominance of *Thalassiosira* spp. and *Pauliella taeniata*. The total phytoplankton abundance and biomass were 1438 × $10^3$ cell/L and 228.21 mgC/m$^3$, respectively.

In the freshwater runoff zone of influence, it is worth mentioning the abundant marine euryhaline cold-water early spring Arctic species *Navicula septentrionalis*, whose abundance exceeded that of *Pauliella taeniata* by 1.8 times. In the water area of the Yenisei river estuary (station 11), another cardinal structural rearrangement of the phytoplankton community was observed. The indicators of the quantitative development of the basic species assemblage dropped sharply: the abundance of *Thalassiosira* spp. dropped by nine

times and that of *Pauliella taeniata* dropped by 30 times. Both in abundance (62%) and biomass (86%), the ice-neritic species *Fossulaphycus arcticus* dominated in the community; an assemblage of small-celled species of the genus *Chaetoceros* occupied a subdominant position. The samples contained many empty frustules and valves of *Aulacoseira* spp., *Diatoma* sp., and *Lindavia comta*.

Stations 12–16 were river stations (EZ), and the phytoplankton community here comprised several freshwater species: *Aulacoseira* spp. (mainly *A. granulata* and *A. italica*), which make up 42% of the total abundance of algae and 75% of their total biomass. *Asterionellopsis formosa* was numerous (27% of the total abundance and 16% of the biomass of phytoplankton). Green algae were constantly encountered with a dominance of *Monoraphidium contortum*, *Pseudopediastrum boryanum*, and *Scenedesmus quadricauda*. The maximum total abundance and biomass of phytoplankton in river water did not exceed $1074 \times 10^3$ cell/L and 125.35 mgC/m$^3$, respectively.

Stage two (13–15 July). After 2 weeks, significant changes occurred in the taxonomic structure of the phytoplankton community, and the leading role in the algae community in the MZ began to be played by heretofore nondominant species. The species complex *Pauliella taeniata, Navicula septentrionalis, Peridiniella catenata, Thalassiosira gravida*, and *Eutreptiella sp.* occupied the top position, in addition to which heterotrophic dinoflagellates *Protoperidinium breve, P. brevipes, P. granii*, and *Gyrodinium lacryma* actively developed. The total phytoplankton abundance at stage two varied within a smaller range of $16–1379 \times 10^3$ cell/L, and the algae biomass varied from 0.04 to 117.22 mg C/m$^3$ (Figure 4). The role of small flagellate algae in the freshwater runoff zone (coastal area) was 0.01–1% of the total phytoplankton biomass. In the MZ, small flagellates accounted for 9–10% of the total phytoplankton biomass, and only at station 22 did their share sharply increase to 95%.

In the EZ (station 17), the species composition of phytoplankton and their numerical abundance were close to those at station 12 at stage one. It is only necessary to note the appearance of an assemblage of brackish-water and marine species, which, rapidly developing, had already completely replaced freshwater forms at station 18. The number of representatives of the genus *Thalassiosira* increased here by 12 times, and that of biomass by 14 times. The number of *Pauliella taeniata* cells exceeded that of *Thalassiosira* spp., which accounted for 58% of the total number of algae, excluding spores. The chrysophycean alga *Dinobryon balticum* also had a relatively high abundance and a low biomass. This species accounted for 29% of the total abundance of algae. The development of *D. balticum* was confined to this part of the water area only. The total phytoplankton abundance and biomass at station 18, excluding spores, were $361 \times 10^3$ cell/L and 78.30 mgC/m$^3$, respectively.

At station 19 in the EZ, phytoplankton mainly contained species of the basic assemblage, and there was a clear trend of a westward decrease in the quantitative development indicators of algae. The abundance of algae at station 20 versus the previous station decreased by 5.3 times, and that of biomass decreased by 4.8 times. The colonial phototrophic dinoflagellates *Peridiniella catenata* accounted for 14% of the abundance and 28% of the biomass; being a constant component of the pelagic phytoplankton community throughout the studied water area, they achieved appreciable development only in this area. In geographical position, station 20 almost completely corresponded to station 9. The similarity of the species composition between them exceeded 60%, while the total phytoplankton abundance was 55 times less, and the biomass 27 times less, than the values recorded here 2 weeks earlier.

The trend of a westward decrease in the abundance of algae persisted up to station 25, west of the Kara Strait. However, dividing cells of *Pauliella taeniata* and *Thalassiosira gravida* were constantly encountered. In this part of the MZ, 9–54% of the abundance of algae and 67–89% of their total biomass comprised heterotrophic dinoflagellates *Protoperidinium breve, P. brevipes, P. pallidum*, and *Gyrodinium lacryma.*

In the water area of the Barents Sea adjacent to the Kara Strait (stations 1 and 25), at both stages, the phytoplankton biomass was extremely low, although the abundance of algae at stage one was commensurate with that in the adjacent area of the Kara Sea.

### 3.4. Primary Production

The Chl *a* concentration in the surface layer of the MZ at stage one varied over a very wide range from 0.28 to 10.36 mg/m³ (Figure 5). The minimum values (0.28–0.88 mg/m³) were in the western part of this zone within the solid ice field; the Chl *a* concentration increased to 3 mg/m³ in the eastern part of the MZ and up to 10.3 mg/m³ at station seven. The same high concentration was found in the FZ. At stage two, the Chl *a* concentration was at the minimum level (0.06–0.27 mg/m³) in almost the entire MZ and increased to 6.7 mg/m³ in the FZ. In the low salinity zone at stations near the FZ, moderate values were determined at both stages (2.62–4.64 mg/m³), and high values (6.68–8.01 mg/m³) in the EZ. The share of pheophytin in the western part of the MZ (stations 1–4 at stage one and stations 21–25 at stage two) averaged 55% of total Chl *a* and pheophytin. In the rest of the study area, the share of pheophytin averaged 24%.

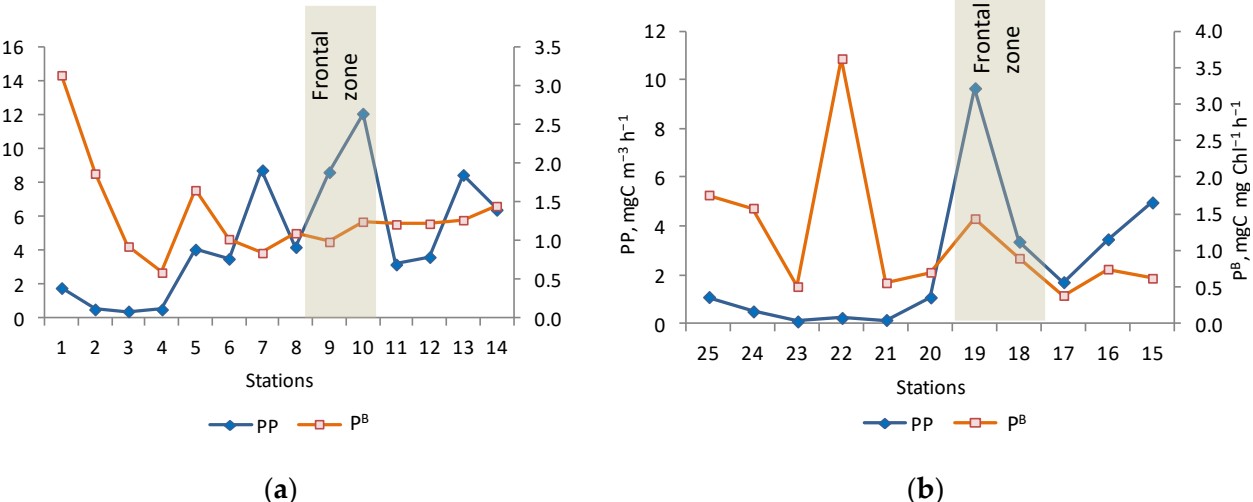

**Figure 5.** Distribution of surface PP and $P^B$ values in the study area at stage 1 (**a**) and stage 2 (**b**).

The PP values at stage one within the MZ varied greatly. In the water area covered with ice (western part of the MZ, stations two–four), the PP values were low (on average, $0.49 \pm 0.07$ mgC m$^{-3}$ h$^{-1}$). In the eastern part of the MZ, it increased to 3.52–8.75 mgC m$^{-3}$ h$^{-1}$ with a maximum at station seven (Figure 5). The PP values at stage two in the MZ were at a low level (0.09–0.50 mgC m$^{-3}$ h$^{-1}$). High PP values were observed in the FZ at both stages (8.64–12.1 mgC m$^{-3}$ h$^{-1}$). In the freshened zone near the FZ (stations 11, 12, and 17), the PP values were relatively low (1.70–3.62 mgC m$^{-3}$ h$^{-1}$), but increased in river water to 8.47 and 4.97 mgC m$^{-3}$ h$^{-1}$ at stages one and two, respectively.

The $P^B$ values at most stations of stage one varied from 0.59 to 1.27 mgC mgChl $a^{-1}$ h$^{-1}$, with a slight upward trend in the direction from the MZ to the EZ, but the maximum values were reached in the western part of the MZ at stations one, two, and five (1.67–3.14 mgC mgChl $a^{-1}$ h$^{-1}$) (Figure 5). At stage two, the $P^B$ values varied greatly in the MZ (from 0.51 to 3.62 mgC mgChl $a^{-1}$ h$^{-1}$), with a maximum at station 22 and, to a much lesser extent, in the FZ and EZ (0.39–1.44 mgC mgChl $a^{-1}$ h$^{-1}$), with a maximum at station 19 in the FZ.

The quantum yield of PSII ($F_v/F_m$) at stage one was highest (0.661–0.729) on surface waters in the EZ, and moderate (0.443–0.597) in the FZ and eastern part of the MZ (Figure 6). In the area of continuous ice cover in the western part of the MZ (stations three and four), the potential photosynthetic activity was at a minimum (0.056–0.312), increasing again towards the Kara Strait. The $F_v/F_m$ values in surface waters at stage two were highest (0.551–0.716) in the EZ and the FZ, lowest (0.125–0.249) in the eastern part of the MZ and moderate (0.450) in the western.

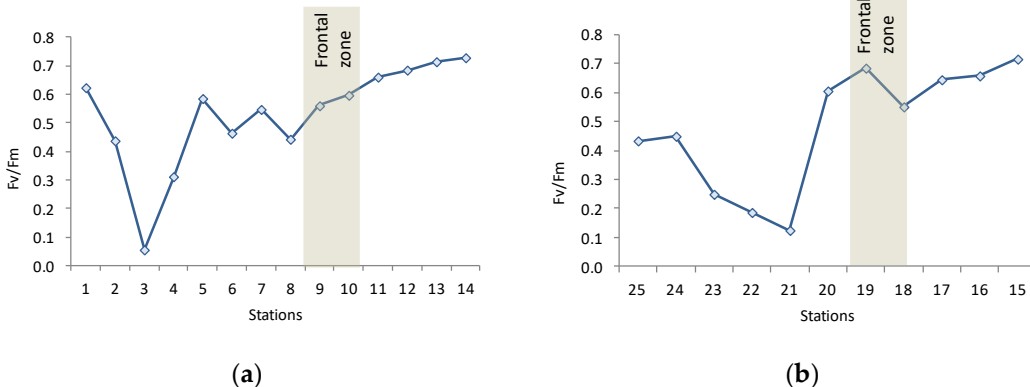

**Figure 6.** Distribution of surface $F_v/F_m$ in the study area at stage 1 (**a**) and stage 2 (**b**).

*3.5. Statistical Analyses*

Diversity. At stage one, the richness and diversity generally increase eastward from the Barents Sea, reaching the peak in the eastern part of the MZ (station seven), then dropping in the EZ. Two weeks later, at stage two, the opposite pattern was observed, with the highest diversity in the EZ (Figure 7).

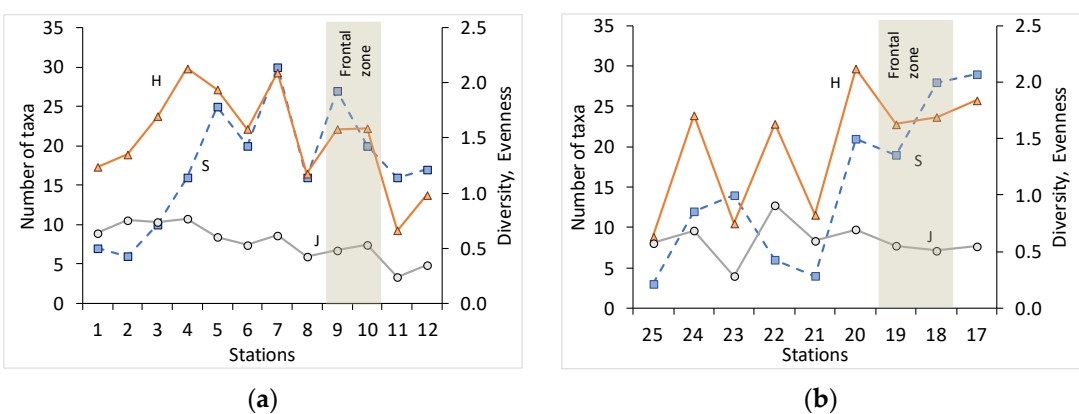

**Figure 7.** Distribution of number of taxa (S), diversity (H), and evenness (J) in the study area at stage 1 (**a**) and stage 2 (**b**).

Community composition. The *n*MDS ordination by community composition of several "marine" stations (stations 1–3 and 21–25) were clearly separated from others located close to the Ob River discharge zone (Figure 8). The significance of the difference between these two groups was confirmed by the one-way ANOSIM test (R = 0.663; *p* <0.001). Plankton of the first group of stations was characterized by a relatively high contribution of flagellates, mainly dinoflagellates *Gyrodinium lacryma*, *Gymnodinium sp.*, and *Protoperidinium pallidum* (on average up to 40% of the total biomass), and an unidentified group of small phototrophic microalgae (26%) and cryptophytic algae *Teleaulax amphioxeia* (10%), while the diatoms (*Thalassiosira spp.*, *Pauliella taeniata*, *Melosira varians,* and *Fossulaphycus arcticus*) constituted up to 70% of the biomass at the stations of the second group.

Three abiotic variables (salinity, silicate, and nitrite concentrations) had the highest correlations with the biotic ordination positions (Figure 8), marking the separation between these two groups of stations (all the correlation coefficients > 0.7, *p* < 0.01). The four stations from the frontal zone (stations 9, 10, 18, and 19) were situated close together, indicating high similarity in their phytoplankton composition.

Impact of environmental factors. In general, nine environmental parameters were responsible for a considerable part of microplankton variability (16–75%, on average about 50%), while different integral characteristics had their own particular combination

of best-fitting predictors (Table 3). The total biomass and phototrophic biomass were mainly determined by phosphate concentration; the photo-to-heterotrophic ratio, Chl *a* and phaeophytin concentrations, PP and potential photosynthetic capacity ($F_v/F_m$) were mainly determined by silicate concentration. The pure effect of differences between stages was statistically significant for PP and the relative phototroph biomass, but this factor's contribution did not exceed 11–16% of total variations. Indeed, while PP was on average twice as high as that at stage one ($4.73 \pm 3.63$ vs. $2.39 \pm 2.90$ mgC m$^{-3}$ h$^{-1}$), this difference was insignificant (Mann–Whitney test, $p = 0.057$). Similarly, the percentage of phototrophic biomass was also insignificantly higher at stage one ($89.4 \pm 20.1$ vs. $64.2 \pm 37.8\%$, $p = 0.113$). Thus, spatial variability predominated over the short-term temporal changes in Protist biomass and production. However, the difference for these parameters separately in the MZ between stages was statistically significant (Mann–Whitney test, $p = 0.003$). Neither water temperature nor dissolved inorganic nitrogen had a considerable effect on the functional characteristics of phytoplankton, but were related to species composition and diversity.

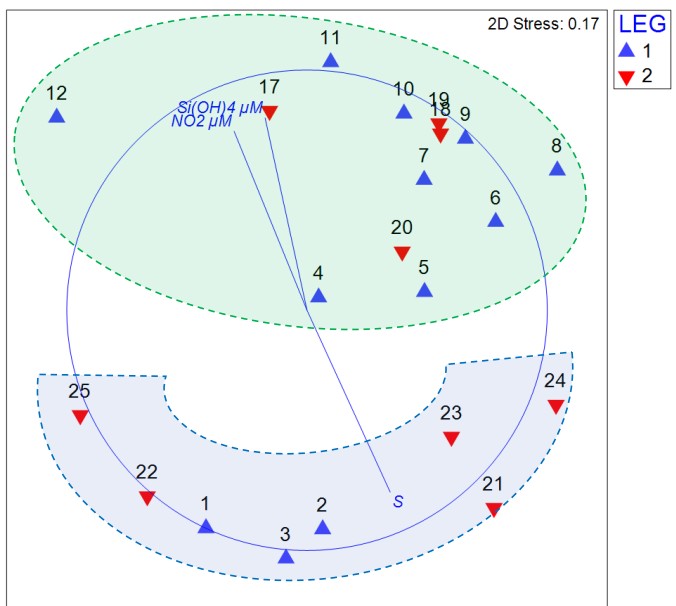

**Figure 8.** *n*MDS ordination plot of stations (numbers). Shaded areas mark the estuarine (upward) and marine (downward) groups of stations. Vectors indicate the environmental variables with the highest correlation with the biotic ordination positions.

**Table 3.** Significance level (*p*) and proportion of explained variation (% Var) for predictor included in the final best-fit model. Significant values are highlighted in bold.

| Abiotic Parameters | | Total Biomass | Biomass Phototrophs | % of Phototrophs | % of Nanoflag. | Chl *a* | Primary Production | $F_v/F_m$ | Species Composition | Richness | Diversity |
|---|---|---|---|---|---|---|---|---|---|---|---|
| | | | | | | **Biotic (Response) Variables** | | | | | |
| Stage | *p* | — | — | **0.046** | — | 0.217 | **0.026** | 0.130 | 0.079 | — | — |
| | % Var | — | — | **16.0** | — | 4.111 | **11.0** | 4.125 | 6.4 | — | — |
| T water | *p* | — | — | — | — | — | — | — | 0.333 | 0.209 | **0.030** |
| | % Var | — | — | — | — | — | — | — | 4.9 | 7.5 | **22.5** |

**Table 3.** *Cont.*

| Abiotic Parameters | | Biotic (Response) Variables | | | | | | | | | |
|---|---|---|---|---|---|---|---|---|---|---|---|
| | | Total Biomass | Biomass Phototrophs | % of Phototrophs | % of Nanoflag. | Chl $a$ | Primary Production | $F_v/F_m$ | Species Composition | Richness | Diversity |
| Salinity | $p$ | — | 0.083 | — | — | 0.195 | **0.038** | 0.240 | 0.387 | **0.047** | 0.324 |
| | % Var | — | 6.6 | — | — | 5.6 | **12.2** | 2.5 | 4.8 | **19.6** | 4.6 |
| Ice | $p$ | — | — | — | — | — | — | **0.032** | 0.178 | — | — |
| | % Var | — | — | — | — | — | — | **8.8** | 5.7 | — | — |
| PO$_4$ | $p$ | **0.001** | **0.001** | — | 0.296 | — | — | — | — | **0.043** | — |
| | % Var | **72.8** | **56.6** | — | 4.4 | — | — | — | — | **13.7** | — |
| Si(OH)$_4$ | $p$ | — | — | **0.019** | **0.032** | **0.001** | **0.003** | **0.001** | **0.017** | 0.244 | — |
| | % Var | — | — | **24.7** | **20.7** | **36.2** | **34.5** | **52.8** | **9.9** | 4.5 | — |
| NO$_2$ | $p$ | — | 0.059 | — | — | — | — | — | **0.048** | — | **0.026** |
| | % Var | — | 7.4 | — | — | — | — | — | **7.5** | — | **18.6** |
| Total % Var | | 72.8 | 75.1 | 40.7 | 31.9 | 45.9 | 57.7 | 68.2 | 39.2 | 56.3 | 45.7 |

## 4. Discussion

### 4.1. Nutrients

The content of nutrients (concentrations of dissolved inorganic nitrogen (DIN) and phosphates) at stage one, i.e., during a period of ice melting, varied widely both in the MZ and EZ of the southwestern Kara Sea. The DIN concentrations at different stations within these zones differed by 43 and 6 times, respectively, and phosphate concentrations differed by 3.5 and 2 times (Table 1). After 2 weeks, the range of variation in nutrient concentrations in both zones decreased significantly, mainly due to a decrease in maximum concentrations. Note that for the silicate concentration, the distribution pattern was similar for both stages, characterized by uniformly high values in the river and EZ and uniformly low values in the MZ (Table 2). The frontal zone, located in the area of stations 8–10 at stage one, shifted eastward to the area of stations 18–20 in 2 weeks, confirmed by data on salinity, nutrients, and phytoplankton and production characteristics of algae.

The phosphate and DIN contents in the MZ in the surface layer decreased markedly within 2 weeks, from values above or close to the limiting level (1.34–7.24 μM) at stage one to values significantly below the limiting level (0.35–0.99 μM) at stage two. At low temperatures, the limiting levels of the main nutrients for PP are 0.5 μM P, 2 μM N, and 2 μM Si [5,41]. The surface content of silicate in these areas remained at a high level throughout the entire research period, significantly exceeding the limiting level.

### 4.2. Chromophoric Dissolved Organic Matter

The average BIX, Sr, and SUVA values in this region are typical of natural waters with the predominance of terrigenous CDOM (see, e.g., the review by Derrien et al. [42]). As previously mentioned, the maximum SUVA value was measured in the Yenisei River estuary at stations 15–18. It was slightly higher than that reported for the Yenisei River end member during the freshet in June 2004 and 2005 (3.10–3.64 m$^2$/gC) [17]. The value of the spectral slope ratio also agrees well with the data reported in [17]. Variation in the SUVA spectral slope is typical of riverine waters and river-impacted ocean margins [43]. With increasing distance from the Yenisei River estuary, SUVA dropped gradually due to an increase in the contribution of autochthonous organic matter and the decomposition of organic matter during photo- and biodegradation (Figure 3). West of the Yamal Peninsula,

the influence of Ob and Yenisei runoff was weak, suggesting that algae and bacteria are the main source of CDOM. The BIX, SUVA, and Sr values are comparable with the data reported for Kara Sea surface waters in autumn 2015 [44] and indicate the mixed riverine/marine DOM nature with a predominant autochthonous component.

### 4.3. Phytoplankton Community

Analysis of the state of phytoplankton in the southwestern Kara Sea in different seasons makes it possible to identify key trends in the development and changes in the structure and functioning of algae, as many authors have discussed [3,6,7,9–12,14,45]. According to the latest ideas, four phases can be distinguished in the annual succession cycle of phytoplankton in the southwestern Kara Sea: the first phase is the cryoflora bloom, which usually occurs in February, dominated be *Amphiprora paludosa f. hyperborea*, *Nitzschia frigida*, and *Thalassionema nitzschioides*. The second phase is the bloom, which normally occurs in April at the edge of melting ice, dominated by *Thalassiosira antarctica var. borealis*, *T. gravida*, *T. hyalina*, *Chaetoceros curvicetus*, and *C. socialis*. The third phase is late spring and summer development (July–August) with a change in dominant assemblage, consisting of the species belonging to the genera *Thalassiosira* and *Aulacoseira*, colonial diatoms (*Pauliella taeniata*, *Navicula septentrionalis*, and *Pseudo-nitzschia seriata f. seriata*), and *Peridiniella catenata*, as well as heterotrophic dinoflagellates and euglenids. The fourth dormant phase begins at the end of September and continues until freezing in October–November. During this period, there is a reduction in the species diversity of algae, as well as in phytoplankton abundance.

Based on the results of our research in the southwestern Kara Sea in July 2018, in the late spring–early summer phytoplankton development during ice melting, the high production potential of algae was realized as a mosaic composed of short-lived and relatively small (in terms of the water area) phytoplankton assemblages, governing mainly by salinity and silicate concentration.

These assemblages differed mostly in number taxa and diversity. The potential photosynthetic activity of phytoplankton (Fv/Fm value) was very high in the freshened EZ, confirmed by data on the phytoplankton abundance in this area (Figures 4 and 6).

Of all the Russian Arctic seas, the Kara, Laptev, and East Siberian are the most similar, for which a greater river runoff governs their characteristics. Our data on the phytoplankton structure, species composition of algae and seasonal development in the southwestern Kara Sea agree well with the results of long-term studies in 1996–2001 [4]. In general, algae species found in Kara Sea phytoplankton are characteristic of all Arctic seas. As for phytoplankton biomass, comparison is not always possible or correct. In almost all publications, the algal biomass is given in chlorophyll units; more rarely, it is given in wet weight (biovolume), and it is given in carbon units in carbon units [7,10,46]. Nevertheless, the first data obtained on the structure of East Siberian Sea phytoplankton in the zone of influence of the Indigirka and Kolyma rivers indicate similar values of the abundance and biomass of algae (in carbon units) compared with similar indicators in the zone of influence of the Ob and Yenisei Rivers in the Kara Sea [47]. Additionally, similar values of the abundance and biomass of algae in carbon units were obtained in the Laptev Sea in the zone of influence of Lena freshwater runoff [48].

### 4.4. Phytoplankton Production

At stage one in the MZ, the potential photosynthetic activity of algae under the ice fields (stations three and four) was at a minimum (0.056 and 0.312, respectively) with a very low level of phytoplankton abundance, while in the rest of the area it was moderately high (0.44–0.62). Two weeks later, the $F_v/F_m$ values and phytoplankton abundance in the MZ decreased to a low level in the central part, remaining moderate only in the area of the Kara Strait. The maximum PP values and maximum or close to maximum values of the phytoplankton abundance and biomass of both, at stage one and 2 weeks later at stage two, were observed in the FZ, several times higher than the PP values in the rest of the study

area (1.5 times at stage one and 3 times at stage two). For 2 weeks between the stages, the PP and phytoplankton values remained quite high in river water (EZ) and moderate in the river–seawater mixing zone, although a decrease in PP by 1.5–2 times was observed (Figure 4). In the water area of the Barents Sea (stations 1 and 25), at both stages, the PP value was two–four times higher than that in the adjacent area of the Kara Sea.

The values of the quantum efficiency of PSII for 2 weeks in the FZ and EZ were at a high level (0.55–0.73), reflecting the consistently high potential photosynthetic activity of phytoplankton in these zones. It should be noted that the maximum values of the phytoplankton's physiological activity were observed in river water. Here, in the EZ at stage two, the Chl *a* concentration increased.

The strongest differences in the PP distribution, like for phytoplankton at stages 1 and 2, were observed west of the Yamal Peninsula in the MZ. At the end of June, PP significantly increased from the minimum values in the western part of the MZ to very high values in the eastern. Two weeks later, in mid-July, the PP values in the entire MZ were at the minimum level. The potential photosynthetic capacity of phytoplankton in the area of the Yamal Peninsula at the beginning of the study was high (0.46–0.55), but after 2 weeks it significantly decreased, reflected by low PP values. In addition to the decrease in PP in this area the phytoplankton abundance, Chl *a* concentration, and phosphate content also decreased. The DIN content in the MZ was high only at stage one under the ice field (stations 3 and 4) and decreased below the limiting value at stage two (stations 23 and 24), when about 25% of the water area in this zone was covered with ice.

In the southwestern Kara Sea in early spring, phytoplankton in the MZ are already in an active state with a high potential photosynthetic capacity, the manifestation of which is restrained by a lack of light under ice conditions. Our earlier studies in 2016 in the same area along the same route in continuous ice cover conditions showed that the nutrient content was above the limiting level everywhere [11]. The potential photosynthetic capacity of phytoplankton in early spring was high (0.4–0.6) both under ice and in open water areas. The PP level was limited by the low level of illumination associated with dense ice cover. Just like in the 2018 study, there was a large variability in the structural and functional parameters of phytoplankton from station to station [11].

As our studies showed, PP was high at the time of ice melting (end of June, stage one) and it significantly decreased 2 weeks later (mid-July, stage two); surface PP was also higher in the EZ. The surface PP values obtained in our studies at stage two were comparable with the results of studies carried out in August of the same year (2.61–5.52 and 34.2–63.4 mgC m$^{-3}$ d$^{-1}$, for the southwest and estuary, respectively) [49]. As other studies have shown [5,50–53], surface PP values in the Kara Sea (August–September) vary from 2.61 to 37.5 mgC m$^{-3}$ d$^{-1}$ in the southwestern region and from 3.2 to 88.0 mgC m$^{-3}$ d$^{-1}$ in the Yenisei estuary and river runoff zone. Our previous studies conducted in April showed that under-ice production was extremely low: 0.1–1.92 mgC m$^{-3}$ d$^{-1}$ in the southwestern region and 0–0.13 mgC m$^{-3}$ d$^{-1}$ in the Yenisei estuary [11]. Studies conducted in August–September in different Arctic seas showed similar surface PP values: 3.7v26.3 mgC m$^{-3}$ d$^{-1}$ in the southeastern Barents Sea and 2.5–16.7 mgC m$^{-3}$ d$^{-1}$ in the Laptev Sea [49,51].

The concentration of surface chlorophyll during our studies at the beginning of summer significantly decreased in the southwestern region over 2 weeks, and almost doubled in the EZ during this period. The measured values in August of the same year were comparable to our results, 0.06–0.22 and 1.53–6.71 mg/m$^3$, respectively [49]. As other studies have shown [50,52,53], Chl *a* values in the Kara Sea (August–September) vary from 0.33 to 1.46 mg/m$^3$ in the southwestern region and from 1.32 to 4.65 mgC/m$^3$ in the Yenisei estuary and river runoff zone. In April, when almost the entire sea area was covered with ice, surface Chl *a* varied from 0.14 to 3.22 mg/m$^3$ in the southwestern region and from 0.11 to 0.71 mg/m$^3$ in the river runoff zone [11]. Studies conducted in August–September in different Arctic seas showed 0.11–0.63 mg/m$^3$ in the southwestern Barents Sea and 0.15–0.71 mg/m$^3$ in the Laptev Sea [49,51].

*4.5. Impact of Environmental Factors*

Of the eight environmental factors considered, a significant impact was revealed for five (water temperature, salinity, and silicates, phosphates, and nitrites concentrations). The most significant factor for the share of phototrophs, the Chl *a* content in water, the rate of PP, and potential photosynthetic capacity ($F_v/F_m$) was salinity and silicate concentration in water, which is determined by its importance for the development of the dominant group of phytoplankton in this area-diatoms. The total phytoplankton biomass largely depended on phosphates, which controlled more than 70% of the variability of this parameter. Note the influence of water temperature and the presence of nitrites on the species diversity of phytoplankton, and the influence of the silicate and nitrite concentrations on species composition. In addition, PP and the relative biomass of phototrophs differed statistically significantly in the MZ at different stages—they were higher at stage one.

**5. Conclusions**

It can be stated that the phytoplankton's abundance and its productivity in the south-western Kara Sea during the 2-week period after ice melting may decrease substantially in the MZ and vary slightly in the EZ. The production potential of algae is realized as a mosaic of a relatively small, in terms of the water area of both zones, short-lived phytoplankton community, while photosynthetic activity decreases very quickly. The leading factors influencing these changes are the silicate content in water and salinity.

**Author Contributions:** Conceptualization, S.A.M. and A.F.S.; methodology, S.A.M., A.F.S., E.I.D., A.N.D. and P.V.K.; validation, S.A.M., A.F.S., E.I.D., A.N.D. and P.V.K.; formal analysis, S.A.M., A.F.S., E.I.D., A.N.D., P.V.K. and A.I.A.; investigation, S.A.M., A.F.S., E.I.D., A.N.D., P.V.K. and N.A.B.; writing—original draft preparation, S.A.M., A.F.S., E.I.D. and A.N.D.; writing—review and editing, S.A.M., A.F.S., A.N.D. and A.I.A.; visualization, S.A.M., A.N.D. and A.I.A.; supervision, S.A.M., A.F.S., P.V.K. and N.A.B.; project administration, A.F.S. All authors have read and agreed to the published version of the manuscript.

**Funding:** This research was supported by the Russian Science Foundation, project no. 22-17-00011.

**Institutional Review Board Statement:** Not applicable.

**Informed Consent Statement:** Not applicable.

**Data Availability Statement:** The data are contained within the article.

**Acknowledgments:** The authors are deeply grateful to Aaron Carpenter for correcting the English translation. The authors are also grateful to the three anonymous reviewers for their thorough work with the manuscript.

**Conflicts of Interest:** The authors declare no conflict of interest.

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
