# Peer review of "Structure and Productivity of the Phytoplankton Community in the Southwestern Kara Sea in Early Summer"

_jmse, doi:10.3390/jmse11040832_

Round 1
Reviewer 1 Report
Comments to manuscript jmse-2286533 entitled “Structure and productivity of the phytocenosis in the southwestern Kara Sea in early summer”
This paper provides a description of the phytoplankton community and environmental variables in the southwestern Kara Sea in early summer 2018. This study contributes to the knowledge on the seasonality of the Arctic marine ecosystem and is important, particularly due to climate-induced rapid changes in the Arctic through rising temperature and declining sea ice cover. However, there are several issues in the manuscript that need to be clarified before the manuscript can be recommended for publication.
My main concern goes to the approach to reveal the main environmental factors effected phytoplankton structure, composition, and productivity, in which any statistical analysis is omitted. The authors claim that "the development of phytoplankton and its physiological state in the southwestern part of the Kara Sea is determined by physical environmental factors: the presence or absence of ice cover, fresh-water runoff, the availability of biogenic elements, and the degree of activity of algae populations" (Abstract: lines 19-25; Conclusions: lines 604-608). However, this statement is not confirmed by results of analysis of the relationships between structural and physiological parameters of the phytoplankton and measured environmental variables. At the same time, the authors mentioned in the section 'Materials and Methods' that correlation analysis was applied to evaluate the relations between environmental variables and phytoplankton, and differences between biological parameters in different sampling sites were tested by an independent sample t-test for equality of the means (lines 209-213).
1. I recommend to include the results of the performed statistical analysis in the section 'Results', and to discuss it in the section 'Discussion' to confirm the authors' conclusions.
2. Since parametric Pearson’s correlation coefficient was selected to assess the linear relationship between the measured environmental variables and phytoplankton characteristics, data normality and homoscedasticity need to be tested before the analyses and this needs to be indicated in the text. Pearson's correlation coefficient is efficient for measuring strength of relationship between normally distributed data. For non-normal data, I would suggest non-parametric correlation methods.
Moreover, if the relationship in question is not linear, then correlational analysis will be unrevealing and multivariate statistical methods would be more appropriate.
3. Since parametric sample t-test for equality of the means was applied to assess a statistically significant difference between the means in different phytoplankton assemblages, normality of the data need to be tested before the analyses as well. For non-normal data within a small dataset (12 and 9 sampling points), I would suggest non-parametric tests.
4. The authors describe the phytoplankton structure in terms of species composition and dominant species/groups, however, the indices of species richness, species diversity, and community evenness were not taken into account to characterize the structure of the phytoplankton community. I would recommend to consider these indices in the analysis.
Further, I would recommend to use the currently accepted names of microalgal species and systematic groups throughout the text and to specify the applied nomenclatural system in the section 'Materials and Methods' (e.g., conventional and continuously updated online resources AlgaeBase, WoRMS) to avoid taxonomic confusion:
Line 299: Navicula vanhoeffenii is currently regarded as a synonym of Navicula septentrionalis Cleve.
Line 301: Dinophyta.
Did you mean the class Dinophyceae?
Line 310: Gyrodinium lachryma
"lacryma" was the original spelling employed by Meunier (1910). The currently accepted taxonomically name of this species is Gyrodinium lacryma (Meunier) Kofoid & Swezy 1921 (see Guiry and Guiry, 2023; Algaebase)
Line 312: Plagioselmis prolonga
Plagioselmis prolonga has been found to be the haploid form of the diploid Teleaulax amphioxeia (Altenburger et al. 2020: Dimorphism in cryptophytes - The case of Teleaulax amphioxeia/Plagioselmis prolonga and its ecological implications. Science Advances. 6. 1-8).
Therefore, this name is currently regarded as a synonym of Teleaulax amphioxeia (W.Conrad) D.R.A.Hill.
Was Teleaulax co-occurring in your samples?
Line 384: Fossula arctica is currently regarded as a synonym of Fossulaphycus arcticus S.Blanco.
Line 386: Cyclotella comta is currently regarded as a synonym of Lindavia comta (Kützing) T.Nakov & al.
Line 395: Pediastrum boryanum is currently regarded as a synonym of Pseudopediastrum boryanum (Turpin) E.Hegewald.
Abstract
Abstract is a self-contained and autonomous summary of a text, but not a word-for-word repetition of sentences from 'Introduction' (lines 30-33; 66-68) and 'Conclusions' (lines 604-610).
Please, rewrite to summarize all the key points of your paper but to avoid copy-pasting sentences from the main text.
Results
The results of this study are presented in descriptive way, particularly in section 3.3 'Structure of the phytoplankton community', in which phytoplankton community was characterized from station to station. This part is too detailed and needs to be shortened and summarized.
I would suggest to apply more analytical than descriptive style when presenting your data by including the results of statistical analysis to support your conclusions.
Discussion
The discussion of the results obtained in the southwestern Kara Sea in early summer needs to be extended by comparison with other Arctic basins and other seasons in more detail.
The present language quality is not good enough and needs to be improved.
When describing the numerical abundance of phytoplankton, I recommend to use scientific normalized exponential notation (× 103) before the units in place of the word 'thousand' throughout the text.
In conclusion, I cannot recommend this manuscript for publication in its existing format. The text should be comprehensively revised and resubmitted for review.
Specific comments:
Line 8: Affiliation
Comment: Please correct numbering in front of the Murmansk Marine Biological Institute.
Lines 19-23; 604-608: We can state that the development of phytoplankton and its physiological state in the southwestern part of the Kara Sea is determined by physical environmental factors: the presence or absence of ice cover, fresh-water runoff, the availability of biogenic elements, and the degree of activity of algae populations.
Lines 556-560: The mosaic pattern of phytocenosis, at least at this time, is determined by physical environmental factors, in particular, the presence or absence of ice cover, freshwater runoff (marine, frontal and estuarine zones), and the degree of activity of microalgae populations.
Comment: I'm not sure if 'activity of algae populations' (physiological activity?) refers to physical environmental factors. Further, the concentration of nutrients is commonly the subject of hydrochemistry.
In the text (lines 91-96), you classify the measured variables into 'hydrophysical (temperature, salinity), hydrochemical (alkalinity, concentrations of nitrite+nitrate, phosphate, silicate, dissolved organic carbon) and hydrobiological (species composition, abundance and biomass phytoplankton, Chl a concentration, chlorophyll fluorescence, and primary production)'.
Please rephrase here and clarify what you mean by 'activity of algae populations'.
Line 24: phytocenosiы
Comment: Remove Cyrillic symbol.
Lines 47-53: The only exception is the article [13], which considers the values of primary production and chlorophyll a during the period of seasonal ice melting in the northwestern Kara Sea. The species composition of algae is not given, except for 6 generic taxa of diatoms predominant in plankton. This article also does not provide data on the concentration of Chl a and measured values of primary production on the sea surface. Only the calculated integral values of these indicators in the water column are given.
Comment: When refer the previous studies, I would recommend to use past tense and 'study/research' in place of the 'article'.
Lines 179-180: The samples were incubated in polycarbonate flasks (50 ml) for 3 hours in the original laboratory incubator
Line 191: Chl a concentration was measured fluorometrically [29].
Line 295: belong to the Bacillariophyta.
Line 356: continued,
Line 357: decreased
Line 545: Thalassiosira and Aulacoseira
Line 547: heterotrophic Dinoflagellates and Euglenoidea
Comment: Italics are inappropriate.
Line 229 and below throughout the text: silicon
Comment: I would suggest to use the term 'silicate' in place of the term 'silicon'
Line 246: The content of dissolved inorganic nitrogen (DIN=NO3+NO2) in the MZ
Table 2: PO4
Comment: Please use subscript numbers in chemical formulas.
Line 253: 3.87-7.24 μA
Comment: please check unit.
Lines 342-349: The leading position in the community is occupied by the Chaetoceros spp. (152 thousand cells/l or 80% of the total abundance of large phytoplankton and 4.46 mg C/m3 or 29% of its total biomass). At the same time, the basic complex of species continues to develop actively, the total number of the community increases 9 times, biomass - 2 times compared to previous stations, the values of the abundance of large phytoplankton are 190 thousand cells/l and 15.61 mg C/m3, respectively. Small flagellates are still abundant; their share in the biomass is 9.2% (1.59 mg C/m3).
Lines 363-365: At st. 8, west of the frontal zone (FZ), the complex of colonial diatoms disappears from plankton, the total number of large algae drops to 129 thousand cells/l, biomass to 19.93 mg C/m3.
Comment: The section 'Results' usually requires the past tense to detail the results obtained. Please correct.
Line 376: Empty shells of Aulacoseira spp.
Comment: I believe the term 'frustule' would be more suitable here.
Line 486: The quantum yield of PSII (Fv/Fm) at stage 1 was highest (0,661-0,729)
Line 491: (0,551-0,716)
Line 492: (0,450)
Comment: Please use a dot (not a comma) as a decimal separator.
Lines 521-522: It was slightly higher than the one reported by Stedmon et al. (2011) for the Yenisei River end member during the freshet in June 2004 and 2005
Comment: Please correct the citation style.
Lines 572-573: For two weeks between the stages, the PP and phytoplankton values remained quite high in river water (RZ) and moderate in the desalinated zone,
Comment: I think that the use of term 'desalinated' is not suitable in this context, because it is the past participle of the verb 'to desalinate' (defined as the remove of salt from sea water), whereas water in river estuaries is low saline due to dilution of sea water with freshwater river inflow.
Please reword.
Line 682: 25. Hobbie, J.E.; Daley, R.J.; Jasper, S. Use of Nuclepore filters for counting bacteria by f luorescence microscopy.
Comment: Misspelling of fluorescence, extra space.
Line 684: 26. Sazhin, A.F.; Artigas, L.F.; Nejstgaard, J.C.; Frischer, M.E. The colonization of two Phaeocystis species
Comment: Please format the generic name 'Phaeocystis' in italics.

Author Response
The authors are grateful to the reviewer for valuable comments that improve the quality of the article.
Reviewer :
1.«My main concern goes to the approach to reveal the main environmental factors effected phytoplankton structure, composition, and productivity, in which any statistical analysis is omitted. The authors claim that "the development of phytoplankton and its physiological state in the southwestern part of the Kara Sea is determined by physical environmental factors: the presence or absence of ice cover, fresh-water runoff, the availability of biogenic elements, and the degree of activity of algae populations" (Abstract: lines 19-25; Conclusions: lines 604-608).
However, this statement is not confirmed by results of analysis of the relationships between
structural and physiological parameters of the phytoplankton and measured environmental
variables. At the same time, the authors mentioned in the section 'Materials and Methods' that
correlation analysis was applied to evaluate the relations between environmental variables and
phytoplankton, and differences between biological parameters in different sampling sites were
tested by an independent sample t-test for equality of the means (lines 209-213). I recommend to include the results of the performed statistical analysis in the section 'Results', and to discuss it in the section 'Discussion' to confirm the authors' conclusions».
Authors: An analysis of the relationship between the structural and physiological parameters of phytoplankton and the measured environmental variables has been carried out. The results are presented in the text of the manuscript in the section «Results». The discuss of the analysis included in the section «Discussion» to confirm our conclusions.
Reviewer :
- «Since parametric Pearson’s correlation coefficient was selected to assess the linear relationship between the measured environmental variables and phytoplankton characteristics, data normality and homoscedasticity need to be tested before the analyses and this needs to be indicated in the text. Pearson's correlation coefficient is efficient for measuring strength of relationship between normally distributed data. For non-normal data, I would suggest non-parametric correlation methods. Moreover, if the relationship in question is not linear, then correlational analysis will be unrevealing and multivariate statistical methods would be more appropriate. Since parametric sample t-test for equality of the means was applied to assess a statistically significant difference between the means in different phytoplankton assemblages, normality of the data need to be tested before the analyses as well. For non-normal data within a small dataset (12 and 9 sampling points), I would suggest non-parametric tests».
Authors: The wishes of the reviewer we have taken into account. All changes are made to the text of the manuscript.
Following your suggestion, we performed some additional statistical analyses to corroborate our findings. As you recommended, we gave preference to non-parametric methods. In particular, we 1) calculated the indices of species richness (number of species/taxa), species diversity (Shannon’ H) and community evenness (Pielou’ J); 2) performed the nonmetric multidimensional scaling (nMDS-plot) to visualize compositional changes in communities among stations and the relationships with main abiotic factors; 3) multivariate non-parametric distance-based regression analysis (DistLM) was used to explore more exactly the relationships between variations in the microplankton characteristics (biomass, production, diversity and composition) and nine environmental parameters.
Reviewer :
- «The authors describe the phytoplankton structure in terms of species composition and
dominant species/groups, however, the indices of species richness, species diversity, and
community evenness were not taken into account to characterize the structure of the
phytoplankton community. I would recommend to consider these indices in the analysis».
Authors: The indicators of species richness, species diversity, and evenness of the community were taken into account in the analysis of the data. Necessary changes were made to the text of the manuscript.
Reviewer: « I would recommend to use the currently accepted names of microalgal species and
systematic groups throughout the text and to specify the applied nomenclatural system in the
section 'Materials and Methods' (e.g., conventional and continuously updated online resources
AlgaeBase, WoRMS) to avoid taxonomic confusion».
Authors: The names of species and systematic groups of algae have been corrected in accordance with the nomenclature system AlgaeBase, WoRMS. All corrections have been made; changes have been made to the text of the manuscript. These online resources are listed under section «Materials and Methods»
Reviewer: «Abstract is a self-contained and autonomous summary of a text, but not a word-for-word repetition of sentences from 'Introduction' (lines 30-33; 66-68) and 'Conclusions' (lines 604-610). Please, rewrite to summarize all the key points of your paper but to avoid copy-pasting sentences from the main text».
Authors: The abstract has been rewritten; the key points of the manuscript are reflected in the text.
Reviewer: «The results of this study are presented in descriptive way, particularly in section 3.3 'Structure of the phytoplankton community', in which phytoplankton community was characterized from station to station. This part is too detailed and needs to be shortened and summarized. I would suggest to apply more analytical than descriptive style when presenting your data by including the results of statistical analysis to support your conclusions».
Authors: The section has been shortened and rewritten in a more analytical style. The results of the statistical analysis of the data are included in the text of the manuscript.
Reviewer: «The discussion of the results obtained in the southwestern Kara Sea in early summer needs to be extended by comparison with other Arctic basins and other seasons in more detail.
The present language quality is not good enough and needs to be improved. When describing the numerical abundance of phytoplankton, I recommend to use scientific normalized exponential notation (× 103) before the units in place of the word 'thousand' throughout the text».
Authors: A comparison of the results of our study with data on other Arctic seas has been made. Additions have been made to the text.
Scientific normalized exponential notation (× 103) before the units in place of the word 'thousand' throughout the text have been made.
Correction of the English language of the manuscript by a native speaker is done.
Reviewer’s Specific comments:
- Comment about belonging to MMBI taken into account, numbering corrected.
- Reviewer: «I'm not sure if 'activity of algae populations' (physiological activity?) refers to physical environmental factors. Further, the concentration of nutrients is commonly the subject of hydrochemistry».
Authors: The phrase has been rewritten; the correction has been made to the text.
- Reviewer: In the text (lines 91-96), you classify the measured variables into 'hydrophysical (temperature, salinity), hydrochemical (alkalinity, concentrations of nitrite+nitrate, phosphate, silicate, dissolved organic carbon) and hydrobiological (species composition, abundance and biomass phytoplankton, Chl a concentration, chlorophyll fluorescence, and primary production). Please rephrase here and clarify what you mean by 'activity of algae populations'.
Authors: The phrase has been rewritten; the correction has been made to the text.
Reviewer: «Line 24, Remove Cyrillic symbol».
Authors: Corrections made.
Reviewer: «Lines 47-53, When refer the previous studies, I would recommend to use past tense and 'study/research' in place of the 'article'».
Authors: Corrections made.
Reviewer: «Lines 179-180: The samples were incubated in polycarbonate flasks (50 ml) for 3 hours in the original laboratory incubator».
Authors: Corrections made.
Reviewer: «Lines 179-180, 191, 295, 356, 357, 345,547 – Italics and other…».
Authors: Corrections made.
Reviewer: «Line 229, I would suggest to use the term 'silicate' in place of the term 'silicon'»
Authors: Corrections made in all text.
Reviewer: «Line 246, Table 2: Please use subscript numbers in chemical formulas».
Authors: Subscript numbers in chemical checked.
Reviewer: «Line 253: please check unit».
Authors: The unit checked.
Reviewer: «Lines 342-349, lines 363-365: The section 'Results' usually requires the past tense to detail the results obtained. Please correct».
Authors: Corrections made to the text used Past Tense.
Reviewer: «I believe the term 'frustule' would be more suitable here».
Authors: The term “shell” has been replaced by the term “frustule”.
Reviewer: «Line 486, 491, 492. Please use a dot (not a comma) as a decimal separator».
Authors: Corrections made in all text.
Reviewer: «Please correct the citation style».
Authors: Citation style changed.
Reviewer: « Lines 572-573. I think that the use of term 'desalinated' is not suitable in this context, because it is the past participle of the verb 'to desalinate' (defined as the remove of salt from sea water), whereas water in river estuaries is low saline due to dilution of sea water with freshwater river inflow. Please reword».
Authors: Corrections made
Reviewer: «Line 682: 25. Misspelling of fluorescence, extra space».
Authors: Corrections made.
Reviewer: «Line 684. Please format the generic name 'Phaeocystis' in italics»
Authors: Corrections made.
On behalf of the authors
Sergey Mosharov

Reviewer 2 Report
Let me congratulate the authors with the publication. Despite of the many investigations of the Shirshov Institute along the Siberian shelf, few publications are available. Therefore this paper is most welcome and I hope that it will be available soon.
However, the manuscript has several difficulties. The English needs to be improved. I could have given examples for the introduction but the format chosen by the journal makes it impossible. In order to help the authors a review manuscript must be send in Word and then track changes can be applied. The journals strategy to send ready formatted manuscripts is of little help for manuscripts that need language improvements.
If you find the old term phytocenosis you know that the author comes from the Russian tradition. It is hardly used and I advise the authors to omit it.
Please check her application of the term value. Often you mean biomass, rate etc. 3.14 is a value, but biomass has no value (an example).
In the literature I read CDOM is colored DOM, not chromophoric.
section 3 of the introduction seems unneccesary.
Work is study, when the water area is freed from ice cover: without ice cover? Only the calculated integral values of these indicators in the water column are give: an example of bad and unclear English.
3.3 is far too long. Please only half the length. Does the reader need to know all these details?
We need undertitles in the discussion
The conclusion is a statement, not a conclusion. These sentences are redundant and inconclusive.
Where do you cite
Hirche et al. 2006: Structure and function of contemporary food webs on Arctic shelves: A panarctic comparison: The pelagic system of the Kara Sea – Communities and components of carbon flow. DOI: 10.1016/j.pocean.2006.09.010
Drits et al. Distribution and grazing of the dominant mesozooplankton species in the Yenisei estuary and adjacent shelf in early summer (July 2016).
Continental Shelf Research
Volume 201, 1 October 2020, 104133
No connection to the southeaster Barents Sea?
Author Response
Reply to reviewer 2
The authors are grateful to the reviewer for valuable comments that improve the quality of the article.
Reviewer: «The English needs to be improved… The journals strategy to send ready formatted manuscripts is of little help for manuscripts that need language improvements».
Authors: Correction of the English language of the manuscript by a native speaker is done.
Reviewer: «If you find the old term phytocenosis you know that the author comes from the Russian tradition. It is hardly used and I advise the authors to omit it».
Authors: The term «phytocenosis» in the text of the manuscript is replaced by the term «phytoplankton community».
Reviewer: «Please check her application of the term value. Often you mean biomass, rate etc. 3.14 is a value, but biomass has no value (an example). In the literature I read CDOM is colored DOM, not Chromophoric».
Authors: Of the 33 publications of recent years, 17 papers use the term «colored», 16 papers use the term «chromophoric». So both terms are valid.
Reviewer: «Section 3 of the introduction seems unnecessary»
Authors: Introduction section text changed
Reviewer: «Work is study, when the water area is freed from ice cover: without ice cover? Only the calculated integral values of these indicators in the water column are given»
Authors: The phrase was rewritten; explanations of how the ice cover was assessed were included in the text.
Reviewer: «3.3 is far too long. Please only half the length. Does the reader need to know all these details? We need undertitles in the discussion»
Authors: Section 3.3. shortened and rewritten. Subtitles inserted in discussion section.
Reviewer: «The conclusion is a statement, not a conclusion. These sentences are redundant and inconclusive».
Authors: The «conclusion section» has been rewritten in accordance with the recommendations of all reviewers.
Reviewer: «Where do you cite»
Authors: What the reviewer meant by this phrase is not clear. We know and cite the article Hirche et. al., 2006. The article Drits et al., 2016 is not directly related to the subject of the presented study.
On behalf of the authors
Sergey Mosharov

Reviewer 3 Report
Dear Authors,
I have read part of your papers and believe that the studies carried out are interesting and give an important contribution of knowledge to the areas you have investigated.
However in the abstract I found this sentence:
"Optical properties of chromophoric dissolved organic matter, species com-position, abundance and biomass of all size groups of autotrophic and heterotrophic phyto-plankton and primary productivity parameters are considered."
So I don't understand how one can make the mistake of talking about autotrophic and heterotrophic phytoplankton!
specific comments:
- Figure 3 represents stage 1 only. Report stage 2 as well.
- In figure 4 stage 2 shows incorrect stations! Thus it is difficult to decipher the related text without the correct figure.
Author Response
Reply to reviewer 3
The authors are grateful to the reviewer for valuable comments that improve the quality of the article.
Reviewer: «In the abstract I found this sentence: "Optical properties of chromophoric dissolved organic matter, species com-position, abundance and biomass of all size groups of autotrophic and heterotrophic phytoplankton and primary productivity parameters are considered." So I don't understand how one can make the mistake of talking about autotrophic and heterotrophic phytoplankton!»
Authors: The abstract lists the parameters that we measured in our work. We were not mistaken in speaking of autotrophic and heterotrophic phytoplankton. Autotrophic (strictly speaking, phototrophic) phytoplankton are algae with chloroplasts (mainly Dinophyta). Heterotrophic phytoplankton are algae (strictly speaking, Protista) that have lost their chloroplasts and feed on organic debris, bacterioplankton, or even suck out the contents of other algae (almost all Dinoflagellate).
Specific comments:
Reviewer: «Figure 3 represents stage 1 only. Report stage 2 as well».
Authors: The figure shows the results of both stage 1 (arts. 01-14) and stage 2 (arts. 15-25). The clarification “at stages 1 and 2” has been added to the figure caption.
Reviewer: «In figure 4 stage 2 shows incorrect stations! Thus it is difficult to decipher the related text without the correct figure».
Authors: In figure 4 (stage 2) technical error fixed - station numbers are displayed correctly
On behalf of the authors
Sergey Mosharov

Round 2
Reviewer 1 Report
Comments to revised version of the manuscript jmse-2286533 entitled “Structure and productivity of the phytocenosis in the southwestern Kara Sea in early summer”
The manuscript has been revised significantly and all confusing issues have been addressed.
The only further suggestion is to add to the list of objectives in the ‘Introduction’ the definition of the main environmental variables governing the phytoplankton variability, as well as to include the main results of statistical analysis in the ‘Conclusion’ and ‘Abstract’ (see below). This is an important outcome of your research, I think.
Please, find below listed minor comments and suggestions.
Specific comments/suggestions:
Lines 77-83:
The aim of this work is to assess phytoplankton abundance and activity in the Kara Sea in early summer in and outside the river runoff zone of influence when the water area is freed from ice cover. The specific objectives were to determine (1) the abundance of phototrophic and heterotrophic planktonic algae, (2) species composition of phytoplankton and the relationship between species, (3) Chl a concentration, and (4) rate of primary production and photosynthetically determined potential photosynthetic capacity of phytoplankton.
Suggestion: to add to the list of objectives the definition of the main environmental variables governing the phytoplankton variability.
Line 180:
Algae nomenclature are given in accordance with AlgaeBase and WoRMS.
Comment: References need to be added here and in the List of references, check the recommended citations on these sites.
Materials and Methods
Comment: Please be consistent when specifying the equipment used (name, manufacture, country) through this section:
LCD-thermometer (HANNA Checktemp-1)
Shimadzu TOC-Vcph carbon analyzer coupled with an SSM-5000A solid sample module
T80 spectrophotometer (PG Instruments)
Fluorat-02-Panorama spectrofluorometer (Lumex Instruments)
Leica DM 5000B luminescence microscope
light Carl Zeiss Axio Imager D1 light microscope – delete ‘light’
HAILEA-100 laboratory cooler (Hailea)
LI-192SA quantum sensor
Lines 201-202:
The of Chl a and phaeophytin a concentrations a were calculated according to [32].
Comment: Delete ‘of’ and ‘a’. The Chl a and phaeophytin a concentrations were calculated according to [32].
Lines 223: (Leg (Stage) - ?
Lines 223-224: silicon
Comment: silicate, as elsewhere through the text.
Line 236: un affected
Comment: unaffected
Line 243: show
Comment: showed
Lines 262-263:
The content of dissolved inorganic nitrogen (DIN=NO3+NO2) in the MZ at stage 1 was in 0.07–4.34 μM,
Comment: delete ‘in’ before values.
Line 274:
salinity zone zone to 0.46–0.91 μM
Comment: delete ‘zone’.
Line 281: ND* - no data
Comment: In Table 1 above, there are no missing data (ND).
Lines 310-312:
Among the small flagellar forms, the most common were Dicrateria rotunda: Plagioselmis prolonga, the haploid form of the diploid Teleaulax amphioxeia; and Pyramimonas spp.
Comment: replace colon and semicolon with commas.
Line 312: Pyramimonas spp.
Line 322: Thalassiosira spp.
Line 335: Chaetoceros spp.
Line 349: Chaetoceros spp.
Line 357: Thalassiosira spp.
Line 365: Thalassiosira spp.
Line 370: Aulacoseira spp., Diatoma sp.
Line 372: Aulacoseira spp.
Line 482: Gymnodinium sp.
Line 484: Thalassiosira spp.
Comment: sp., spp. – Italics is not applicable.
Lines 369-362:
In the freshwater runoff zone of influence, it is worth mentioning the numerous marine euryhaline cold-water early spring Arctic species Navicula septentrionalis, the abundance of which exceeded that of Pauliella taeniata by 1.8 times.
Suggestion: In the freshwater runoff zone of influence, it is worth mentioning the abundant marine euryhaline cold-water early spring Arctic species, Navicula septentrionalis, whose abundance exceeded that of Pauliella taeniata by 1.8 times.
Lines 382-383:
Pauliella taeniata, Navicula septentrionalis, Peridiniella catenata, Thalassiosira gravida, Eutreptiella sp. occupied first place,
Suggestion: [species] occupied the top ranks/positions, or [the species complex] occupied first place. Please rephrase.
Line 388-390:
In the MZ, small flagellates of the latter accounted for 9–10% of the total phytoplankton biomass, and only at station 22 did their share sharply increase to 95%.
Comment: of the latter?
Lines 391-391:
In the EZ (station 17), the species composition of phytoplankton and quantitative indicators of its abundance were close to those at station 12 at stage 1.
Suggestion: In the EZ (station 17), the species composition of phytoplankton and its numerical abundance were close to those at station 12 at stage 1.
Line 398: The Chrysophyceae alga Dinobryon balticum
Suggestion: The chrysophycean alga Dinobryon balticum
Line 372-372: Fig. 7
Comment: Please mark ‘Frontal zone’ as in Figs 2, 4-6.
Line 507, 509: Stage 1
Comment: lowercase, stage 1
Line 513: significant. (Mann-Whitney test, p = 0.003).
Comment: delete extra dot before brackets
Line 517: Table 3: PO4, Si(ОН)4, NO2
Comment: subscript
Line 552: biodegradation (Fig. 3)
Comment: add dot at the end of sentence.
Lines 552-554:
West of the Yamal Peninsula, the influence of Ob and Yenisei runoff was weak, that suggesting that algae and bacteria are the main source of CDOM.
Suggestion: West of the Yamal Peninsula, the influence of Ob and Yenisei runoff was weak, suggesting that …
Lines 567-572:
The third phase is late spring and summer development (July-August) with a change in dominant forms, the main of which consist of an assemblage of representatives of the genera Thalassiosira and Aulacoseira, as well as colonial forms (Pauliella taeniata, Navicula septentrionalis, Pseudo-nitzschia seriata f. seriata), and Peridiniella catenata, as well as heterotrophic dinoflagellates and Euglenoidea.
Suggestion:
The third phase is late spring and summer development (July-August) with a change in dominant assemblage, consisting of the species belonging to the genera Thalassiosira and Aulacoseira, colonial diatoms (Pauliella taeniata, Navicula septentrionalis, Pseudo-nitzschia seriata f. seriata), Peridiniella catenata, as well as heterotrophic dinoflagellates and euglenids.
Lines 575-583:
Based on the results of our research, we can conclude that during the period of ice melting, during the late spring – early summer development of phytoplankton in July 2018, in the southwestern Kara Sea, the high production potential of algae is realized as a mosaic of a relatively small, terms of water area, short-lived phytoplankton community. Phytoplankton communities are characterized by a different structure and dominant algae species within a single annual succession cycle. Thus, the potential photosynthetic activity of phytoplankton (Fv/Fm value) was very high in the freshened EZ at both stages, confirmed by data on the phytoplankton abundance in this area (Figs. 6, 4).
Comment: This paragraph is not clear and complicated since it is inconsistent itself. The term ‘mosaic’ means a set of elements (in your case, these are the phytoplankton assemblages) that are somewhat different from each other. What was the difference(s)? Were they differing in terms of production, biomass, abundance, composition, etc.? What environmental variables contributed to these differences? What was the main difference resulting in different production potential? The reference to ‘one year cycle of succession’ is unclear here as the authors’ study was limited to approximately 2 weeks.
Since this is the main authors’ conclusion, I would recommend to clarify it in more detail.
Suggestion:
Based on the results of our research in the southwestern Kara Sea in July 2018, in the late spring – early summer phytoplankton development during ice melting, the high production potential of algae was realized as a mosaic composed of short-lived and relatively small (in terms of the water area) phytoplankton assemblages, governing mainly by .... [environmental variables]. These assemblages differed/altered mostly in … The potential photosynthetic activity of phytoplankton (Fv/Fm value) was very high in the freshened EZ, confirmed by data on the phytoplankton abundance in this area (Figs. 6, 4).
Please rephrase here and in ‘Abstract’ and ‘Introduction’ accordingly.
Lines 584-585:
Of all the Russian Arctic seas, the Kara, Laptev and East Siberian are the most similar, for which a greater river runoff governs their features and characteristics.
Comment: What is the difference between features and characteristics? I would recommend to delete one of these synonyms.
Line 586-588:
Our data on the phytoplankton structure, species composition of algae and seasonal development in the southwestern Kara Sea agree well with the results of long-term studies by different authors in 1996–2001. [4].
Comment: Please delete extra dot. If studies by ‘different authors’ are referred here, why is only one reference [4] cited? Moreover, this reference is included in incomplete format in the List of references.
Please correct this inconsistency and add the authors, journal, volume, and pages to the bibliography.
Hirche et al. 2006: Structure and function of contemporary food webs on Arctic shelves: A panarctic comparison: The pelagic system of the Kara Sea – Communities and components of carbon flow. DOI: 10.1016/j.pocean.2006.09.010

Reviewer 2 Report
The paper has been significantly improved and all recommendations of mine were adequate considered
Author Response
The authors are grateful to the reviewer for valuable comments that improve the quality of the article.